# SoftPose: Learning Soft Attention for Interaction-Aware Multi-Person Image Generation

## Abstract

Pose-guided human image generation aims to synthesize images of individuals performing specific actions based on pose conditions and textual descriptions. While current methods achieve promising results in single-person scenarios, they often struggle to generalize to multi-person settings, particularly under complex spatial interactions. Existing methods typically employ pose guidance in an undifferentiated manner across the image, leading to structural ambiguity and frequent limb entanglement in interaction zones. To tackle this challenge, we propose SoftPose, a novel approach that learns interaction-aware soft attention to adaptively modulate attention flow across pose regions, enabling more fine-grained focus on both self-related and cross-person occlusion areas. By modeling long-range, global spatial dependencies within and across pose regions, SoftPose effectively resolves ambiguities in interactive scenarios while preserving precise single-person pose fidelity. Additionally, we introduce a progressive feature injection strategy that balances global spatial coherence and local pose details across multiple scales. Extensive experiments demonstrate the superiority of SoftPose compared to current methods in generating high-quality multi-person images with complex interactions and varying scenes.

## 1 Introduction

Human image generation aims to synthesize realistic human images under diverse conditions and has wide applications in animation, gaming, e-commerce and human-robot interaction. Although recent text-to-image models can produce visually and semantically coherent images, textual descriptions alone are often insufficient to specify accurate body poses and interactions. Among various pose representations, skeleton-based poses, defined by a sparse set of joint keypoints, offer a compact, interpretable, and computationally efficient format, making them particularly suitable for flexible and real-time human image generation.

To incorporate pose into generation process, current studies commonly adopt a dedicated pose encoder to inject pose features into the image synthesis pipeline. These methods Lu et al. (2024); Ju et al. (2023b); Wang et al. (2024); Yin et al. (2025) have demonstrated strong performance in single-person settings. However, extending them to multi-person scenarios remains challenging. In such cases, skeletons often exhibit overlapping regions, where explicitly modeling the complex spatial interactions and occlusions among individuals become critical. Lacking the ability to capture these inter-person relationships, methods designed for isolated poses often lead to notable inaccuracies, such as entangled or inconsistent structures in interactive regions. Although recent works Kong et al. (2024); Kim et al. (2025) have taken a step forward toward multi-person image generation, some primarily focus on identity preservation, while others rely on costly annotations and lack demonstrations in scenarios with tightly entangled poses and larger numbers of interacting subjects.

To address these challenges, we propose SoftPose, a framework for flexible and robust pose-guided multi-person image generation. SoftPose improves inter-person spatial modeling and pose consistency through two key innovations. First, we design an interaction-aware soft attention mechanism that learns to dynamically reallocate attention weights. We achieve this by computing the overlap areas of dilated skeleton masks, which serve as a prior to derive a soft attention map. This modulated

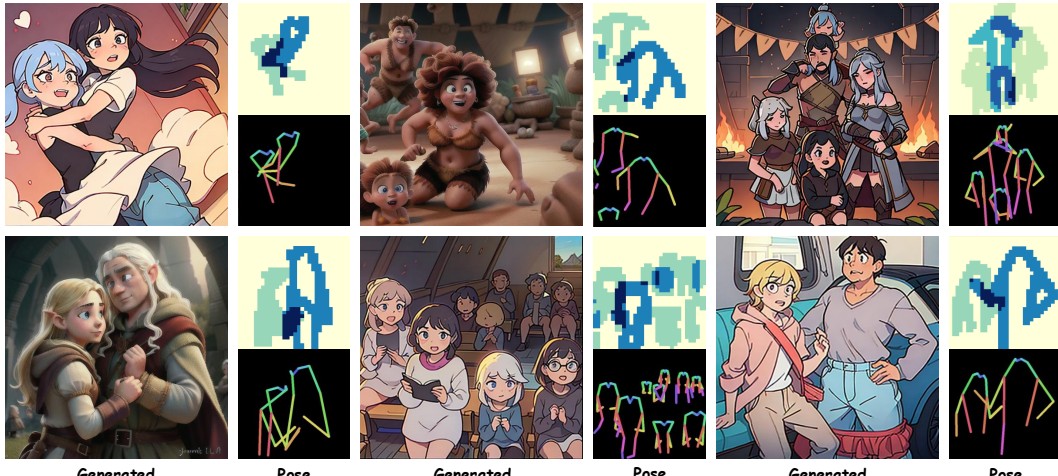

Figure 1: SoftPose generates high-quality multi-person images by modulating attention at overlapping regions (dark blue), ensuring both pose accuracy and natural interactions across scenes.

map assigns higher weights to densely interacting and self-related areas. By seamlessly embedding this mechanism into the T2I pipeline, our method explicitly models inter-person dependencies, enabling the pose encoder to prioritize occluded and intertwined regions. Second, motivated by the hierarchical nature of image generation, we propose a progressive multi-scale feature injection strategy. Instead of feeding all pose features at once, this strategy gradually incorporates features from coarse to fine scales during the training process. This allows the network to first establish the global layout and then incrementally refine fine-grained details, particularly in complex interaction areas. This progressive formulation effectively balances global coherence with local precision, leading to fewer artifacts and enhanced pose fidelity. We conduct extensive experiments on three datasets, demonstrating our method's superior performance in pose preservation and its strong generalization ability, especially in challenging multi-person scenarios with intricate interactions.

This work presents the following key contributions:

- We propose SoftPose, a novel method for pose-guided multi-human image generation that effectively addresses the challenges of complex inter-person interactions and pose consistency in crowded scenes by explicitly modeling inter-person spatial dependencies.
- SoftPose introduces two key innovations: an Interaction-Aware Soft Attention Module that explicitly emphasizes interaction regions to better capture intricate spatial dependencies, and a Progressive Feature Injection Strategy that gradually refines both global layout and fine-grained local details for improved visual fidelity.
- Our extensive quantitative and qualitative experiments demonstrate SoftPose's state-of-the-art performance in pose preservation and its strong generalization ability, especially in highly challenging scenarios with dense subject interactions and complex occlusions.

## 2 RELATED WORK

**Controllable Diffusion Models.** To improve spatial precision and structural alignment in text-to-image generation, recent diffusion-based models Zhang et al. (2023a); Huang et al. (2023) introduce external control signals beyond text prompts. ControlNet Zhang et al. (2023b) pioneered a dual-branch architecture that encodes conditions alongside a frozen base model, enabling fine-grained, structure-aware synthesis. Subsequent work like T2I-Adapter Mou et al. (2024) introduced lightweight adapter modules, offering a cost-efficient way to guide generation without modifying the base model. GLIGEN Li et al. (2023) uses gated self-attention to incorporate layout constraints, while Uni-ControlNet Zhao et al. (2023) extends this by supporting multi-type control through unified adapters at multiple resolutions. A distinct line of research, such as Readout Guidance Luo et al. (2024) directly manipulates intermediate representations during sampling. These methods collectively enhance controllability while maintaining the high-quality output of diffusion models.

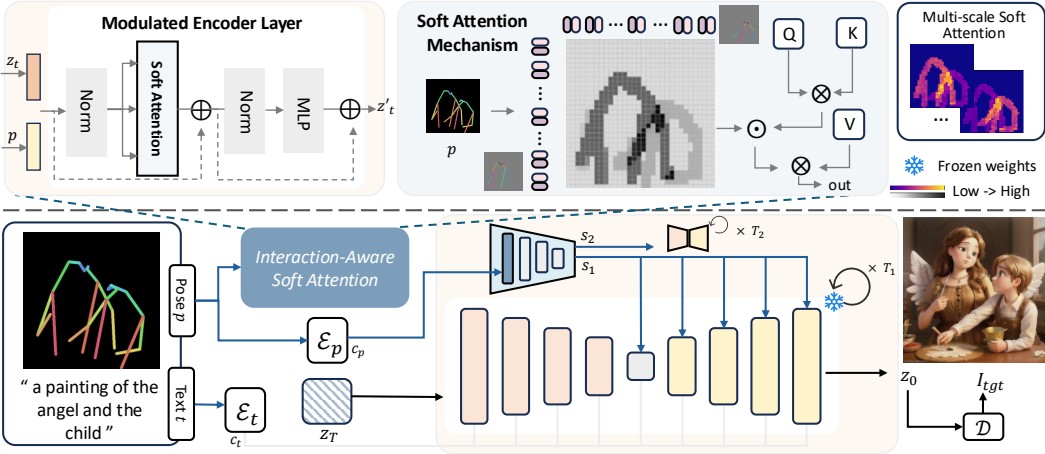

Figure 2: Overview of SoftPose. (1) The upper part illustrates the details of Interaction-Aware Attention Module, which enhances cross-person spatial interactions modeling by refining attention maps with pose masks. (2) The lower part outlines overall denoising process, where the modulated pose guidance progressively guides multi-person image generation.

**Pose-guided Human Image Generation.** Pose-guided human image generation (PGHIG) was first introduced by Ma et al. (2017), who framed the task as generating human images under specific poses using adversarial methods. With the advent of diffusion models, several approaches have improved the fidelity and controllability of PGHIG. PIDM Bhunia et al. (2023) and PoCoLD Han et al. (2023) use cross-attention to align source appearance with target pose, while PCDMs Shen et al. (2023) leverage a multi-stage pipeline for progressive motion and structure modeling. CFLD Lu et al. (2024) introduces a coarse-to-fine latent diffusion strategy, decoupling pose and appearance control using learnable prompts without image-text pairs. HumanSD Ju et al. (2023b) incorporates pose heatmaps in full-model fine-tuning, and StablePose Wang et al. (2024) proposes a ViT-based adapter with hierarchical attention masking for refined pose guidance. GRPose Yin et al. (2025) enhances spatial reasoning by integrating graph structures with diffusion latent spaces. While these methods excel in single-person generation, multi-human synthesis remains underexplored. PersonaCraft Kim et al. (2025) combines diffusion with 3D human modeling for occlusion-aware multi-person generation. Nevertheless, we prioritize robust multi-person generation in more challenging conditions, specifically involving complex occlusions, intricate interactions, and larger group sizes.

## 3 METHODOLOGY

### 3.1 OVERVIEW

Figure 2 shows the architecture of our proposed method. Given the pose control condition $p$ that encodes the skeletal layout of multiple human subjects within a shared image space, and a descriptive text prompt $t$ specifying the scene context and individual appearance attributes, our objective is to generate a high-quality image $I_{tgt}$ in which all $N$ subjects are realistically depicted performing their respective actions in a coherent spatial layout. Our generation pipeline is built on the Stable Diffusion framwrok, as illustrated in the lower part of the figure. It employs a variational autoencoder (VAE) to map images into and out of a compressed latent space $\mathcal{Z}$, and a U-Net denoising network that progressively refines a Gaussian-noised latent $z_T \sim \mathcal{N}(0, I)$ into a clean sample $z_0$ over $T$ timesteps. At each timestep $t \in [1, T]$, the network $\epsilon_\theta(z_t, t, c)$ predicts the noise component $\epsilon$ from the noisy latent $z_t$, where $c$ denotes the conditioning inputs, including the text embedding and pose guidance. This process is optimized by minimizing the standard denoising objective:

$$\mathcal{L} = \mathbb{E}_{z_0, \epsilon, t} \left[ \|\epsilon - \epsilon_\theta(z_t, t, c)\|_2^2 \right], \tag{1}$$

To incorporate semantic information, the input text prompt $t$ is encoded into contextual embeddings $c_t$ via a CLIP text encoder and injected into the U-Net through cross-attention layers. In parallel, the pose control condition $p$ is processed via the Interaction-Aware Soft Attention Module (IA-SAM),

---

**Algorithm 1** Computation of Interaction-Aware Attention Prior

---

1: **procedure** ATTNPRIORCOMPOSER($\{m_i\}_{i=1}^N, h, w$)
2:     $m_i \in \{0,1\}^L$: Binary patch masks for each person
3:     $L \leftarrow h \times w$
4:     Stack masks: $S \leftarrow [m_1; m_2; \ldots; m_N]^\top \in \{0,1\}^{N \times L}$
5:     $o \leftarrow \sum_{i=1}^N S[i,:] \in \mathbb{N}^L$                            ▷ overlap count per patch
6:     $a \leftarrow \mathbb{I}(o > 1) \in \{0,1\}^L$                       ▷ binary overlap indicator
7:     $A_m \leftarrow \mathbf{0}_{L \times L}$
8:     **for** $i = 1$ to $N$ **do**
9:         $A_m \leftarrow A_m + \alpha \cdot (m_i \otimes m_i)$               ▷ self-attention
10:         **for** $j = 1$ to $N, j \neq i$ **do**
11:             $A_m \leftarrow A_m + \beta \cdot \big((m_i \otimes m_j) \odot (a \otimes a)\big)$   ▷ cross-attention (overlap-gated)
12:         **end for**
13:     **end for**
14:     $A_m \leftarrow A_m + \gamma \cdot (a \otimes a)$                   ▷ global overlap boost
15:     **return** $A_m$
16: **end procedure**

---

while a lightweight pose encoder simultaneously supplies sparse features $c_p$. This ensures robustness and enables early-stage feature extraction to be grounded in both structural and relational priors.

Furthermore, to align with the coarse-to-fine nature of the denoising process, multi-scale pose features $s_1, s_2$ from masks of varying semantic granularity are injected during different phases of denoising. Coarse-grained features $s_1$ are applied during the first $T_1$ timesteps to guide global layout, while fine-grained features $s_2$ are injected during the subsequent $T_2 = T - T_1$ timesteps to refine local pose details. This scheduling ensures that spatial guidance evolves progressively from holistic structure to fine articulation, in harmony with the diffusion model's inherent refinement trajectory.

## 3.2 INTERACTION-AWARE SOFT ATTENTION MODULE

The upper part of Figure 2 illustrates the detailed design of Interaction-Aware Soft Attention Module (IA-SAM), which is designed to explicitly model inter-person spatial dependencies. It comprises three core steps: (1) generating an interaction-aware attention prior, (2) modulating the attention computation, and (3) injecting the enhanced pose features into the denoising process.

Given an input pose condition $p \in \mathbb{R}^{N \times 17 \times 2}$, which encodes the 2D coordinates of 17 keypoints for $N$ subjects, IA-SAM begins by generating a spatial presence mask $M_i \in \mathbb{R}^{H \times W}$ for each subject $i$. This is achieved by rendering the skeleton into a binary map, applying morphological dilation and smoothing the result with a Gaussian kernel to obtain a continuous representation of body extent. Each individual's mask is then partitioned into non-overlapping patches, producing binary indicators $m_i \in \{0,1\}^L (L = h \times w)$ that mark skeletal occupancy at the patch level. These patch maps are further processed by the Attention Prior Composer function as defined in Algorithm 1 to synthesize a soft attention prior $A_m$, which encodes spatial relationships essential for multi-person generation.

The attention prior $A_m$ is constructed by composing three semantically distinct attention components, each modulated by a learnable weight that governs its relative influence. The self-attention weight $\alpha$ controls the degree to which each subject attends to its own structural regions, realized as $\alpha \cdot (m_i \otimes m_i)$, promoting intra-person anatomical coherence and reducing limb fragmentation. The cross-attention weight $\beta$ activates attention between subjects only within overlapping zones, implemented as $\beta \cdot ((m_i \otimes m_j) \odot (a \otimes a))$, where $a \in \{0,1\}^L$ is a binary indicator of regions with at least two overlapping subjects. This operation acts as a spatial gate: it retains cross-person attention only in areas where multiple bodies intersect, while suppressing it elsewhere to prevent spurious correlations. The overlap-boost weight $\gamma$ globally amplifies attention responses within regions where multiple subjects intersect, computed as $\gamma \cdot (a \otimes a)$, helping the model resolve ambiguities in densely entangled poses by sharpening focus on high-conflict zones such as limb crossings or contact points.

Background regions are suppressed to near-zero values, ensuring computational resources are concentrated on semantically meaningful areas. The design above is grounded in a key insight from previous studies Tang et al. (2023); Hertz et al. (2022); Wang et al. (2025) that attention maps encode the attribution relationship between conditions and output regions, determining how condi-

tioning signals influence the generated content. By modulating these maps with $A_m$, we directly steer the model's focus toward structurally and interactively critical zones. The attention prior is integrated into a ViT-based layer via multiplicative reweighting of the normalized attention scores. Specifically, given standard attention $\text{Attn}_{\text{std}} = \text{Softmax}\left(\frac{QK^\top}{\sqrt{d}}\right)$, the interaction-aware attention map is computed as:

$$\text{Attn}_{\text{IA}} = \frac{\text{Attn}_{\text{std}} \odot A_m}{\sum_j (\text{Attn}_{\text{std}} \odot A_m)_{ij} + \epsilon}, \tag{2}$$

where $\odot$ denotes element-wise multiplication, and $\epsilon$ is a small constant for numerical stability. The final output of the interaction-aware attention layer is computed as: $Output_{IA} = Attn_{IA} \cdot V$. This formulation allows $A_m$ to act as a spatially adaptive gain controller, guiding the model to focus its representational capacity precisely where pose structure and interpersonal interaction demand the highest fidelity, as depicted in Figure 3.

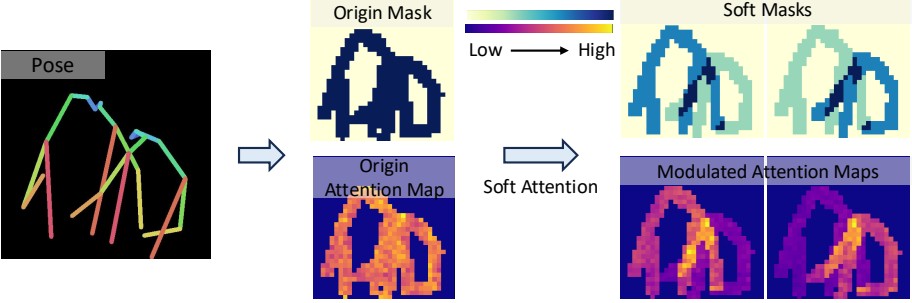

Figure 3: Visualization of original and modulated soft attention maps guided by pose masks, highlighting how IA-SAM enhances cross-person interactions and refines attention focus.

To embed this mechanism into the generative pipeline, we replace the first convolutional layer of the ControlNet encoder with the customized layer incorporating IA-SAM. This ensures that pose features are interaction-aware from the earliest stage of encoding. As denoising progresses, these enhanced features progressively guide the refinement, resulting in images with improved anatomical accuracy, plausible multi-person interactions, and minimal limb-merging artifacts.

### 3.3 PROGRESSIVE FEATURE INJECTION STRATEGY

The dilated masks used to construct the attention prior exhibit different granularity levels depending on the dilation kernel size. Larger kernel sizes lead to broader skeleton coverage, emphasizing coarse spatial structures such as overall human contours and occlusion regions. In contrast, smaller kernel sizes preserve fine skeletal details, better capturing local part-specific features. To leverage this scale-dependent behavior, we design a progressive control strategy that aligns with the inherent coarse-to-fine nature of the diffusion process Cao et al. (2023); Luo et al. (2023); Bao et al. (2023). Specifically, we inject pose features derived from attention priors computed at two dilation scales, scheduled according to timestep:

$$k(t) = \begin{cases} k_1 & \text{if } t > T_2, \\ k_2 & \text{if } t \le T_2, \end{cases} \tag{3}$$

where $k_1 > k_2$ denote the large and small kernel sizes, and $T_2 = T - T_1$ defines the transition point. During early denoising steps ($t > T_2$), large-kernel priors guide the formation of coarse human layouts and occlusion reasoning. In later steps ($t \le T_2$), fine-scale priors refine structural details such as limb geometry and pose articulation. This progressive injection not only enhances the generation of fine-grained body details but also helps to resolve spatial ambiguities and conflicts in interaction regions by explicitly modeling human spatial relationships across scales.

In parallel, to further emphasizes human-centric and interaction-aware regions during training, we augment the base diffusion loss defined in Equation 1 with spatially adaptive weighting. We first define the union of all individual pose masks as $M$, and apply a foreground boost via multiplication:

$$\mathcal{L}_{\text{pose}} = \lambda \cdot \mathcal{L} \cdot M + \mathcal{L} \cdot (1 - M), \tag{4}$$

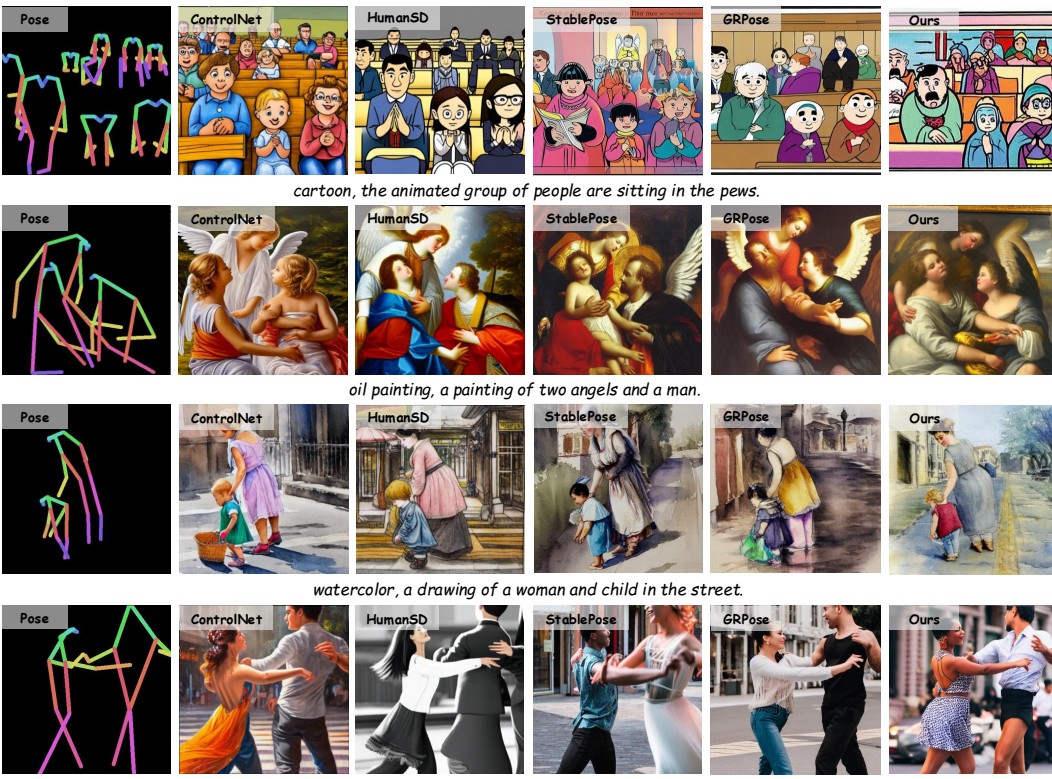

Figure 4: Qualitative comparison between our method and existing baselines on the Human-Art dataset. Each row presents the input pose and prompt, followed by the generated images from different methods.

where $\lambda > 1$ emphasizes the foreground region. To further incorporate fine-grained supervision across multiple individuals, we define the final loss as:

$$\mathcal{L}_{\text{IA}} = \mathcal{L}_{\text{pose}} + \lambda_{\text{multi}} \cdot \frac{1}{N} \sum_{k=1}^{N} (\mathcal{L} \odot M_k), \tag{5}$$

where $\{M_k\}_{k=1}^{N}$ are person-wise masks and $\lambda_{\text{multi}}$ controls the strength of individual constraints.

## 4 EXPERIMENTS

### 4.1 EXPERIMENTAL SETUP

**Dataset.** We follow the experimental settings of Wang et al. (2024); Yin et al. (2025) to conduct experiments on the Human-Art Ju et al. (2023a) dataset. This dataset consists of 50k high-quality images in both natural and artificial scenes with human-centric annotations. Additionally, we evaluate our method on MHP v2.0 Zhao et al. (2020) dataset, which contains 25k images, each with at least two persons. We also use MPII Andriluka et al. (2014) dataset, which includes approximately 25k images, containing both single and multi-person scenes. From this, we select 16k samples with accurate action annotations. For both Human-Art and MHP, we adopt the official train-validation splits recommended by the respective authors, while MPII is split following the same ratio.

**Implementation Details.** Our model is built upon the widely adopted Stable Diffusion 1.5 backbone, closely following the settings of previous state-of-the-art works Ju et al. (2023b); Wang et al. (2024); Yin et al. (2025). We train all models using the Adam optimizer Kingma & Ba (2014) with an initial learning rate of 1e-5 for 10 epochs on 4 NVIDIA GeForce RTX 4090 GPUs each featuring 24GB. The per-GPU batch size is set to 1 with gradient accumulation steps of 4 to stabilize training. For the proposed IA-SAM, we initialize all learnable attention weights from a uniform distribution

Table 1: Results on the Human-Art dataset. The best and second-best results are marked in bold and underline, respectively. Note that methods above and below the horizontal line are compared based on their officially released results, which are trained for different numbers of epochs.

| Dataset | Method | Pose Accuracy | | | Image Quality | | T2I Alignment |
|---|---|---|---|---|---|---|---|
| | | AP ↑ | CAP ↑ | PCE ↓ | FID ↓ | KID ↓ | CLIP-score ↑ |
| Human-Art | SD Rombach et al. (2022) | 0.24 | 55.71 | 2.30 | 11.53 | 3.36 | **33.33** |
| | T2I-Adapter Mou et al. (2024) | 27.22 | 65.65 | 1.75 | 11.92 | 2.73 | 33.27 |
| | ControlNet (Zhang et al. 2023b) | 39.52 | 69.19 | 1.54 | 11.01 | **2.23** | 32.65 |
| | Uni-ControlNet Zhao et al. (2023) | 41.94 | 69.32 | 1.48 | 14.63 | 2.30 | 32.51 |
| | HumanSD Ju et al. (2023b) | 44.57 | 69.68 | **1.37** | **10.03** | 2.70 | 32.24 |
| | StablPose Ju et al. (2023b) | 48.88 | 70.83 | 1.50 | 11.12 | 2.35 | 32.60 |
| | **SoftPose (Ours)** | **52.17** | **72.69** | 1.47 | 12.09 | 2.34 | 32.50 |
| | GRPose Yin et al. (2025) | 49.50 | 70.84 | **1.43** | 13.76 | 2.53 | 32.31 |
| | **SoftPose (Ours)** | **56.31** | **73.19** | 1.50 | 13.07 | 2.36 | 32.58 |

centered at values that preserve the pretrained attention behavior, enabling stable gradient flow while allowing data-driven adaptation during training. Two-level pose masks are constructed using Gaussian filters with kernel sizes of 23 and 13, applied respectively during the early and late timesteps of the denoising process. During inference, we adopt a DDIM sampler Song et al. (2020) with 50 denoising steps. Classifier-free guidance is applied with a scale of 7.5, and no conditioning is dropped during training to preserve pose fidelity. To ensure reproducibility, all experiments were conducted with a fixed random seed of 42.

**Evaluation Metrics.** We adopt a multi-faceted evaluation protocol to assess performance along three core dimensions of this task: pose fidelity, image quality, and semantic alignment. Pose accuracy is measured using three metrics: Average Precision (AP), computed following the COCO keypoint evaluation protocol based on Object Keypoint Similarity (OKS) and reported mean AP over OKS thresholds from 0.50 to 0.95; Pose Cosine Similarity-based AP (CAP) focuses on pose similarity by comparing the cosine similarity of poses; and People Count Error (PCE) evaluates the ability to maintain a consistent number of persons. Image quality is evaluated using two standard metrics: Fréchet Inception Distance (FID) Heusel et al. (2017) assesses the quality of the generated images by comparing their distribution to real images; Kernel Inception Distance (KID) Bińkowski et al. (2018) similarly measures the diversity and fidelity of the generated images. Finally, we assess text-image consistency using the CLIP-Score Radford et al. (2021), which calculates the cosine similarity between text and image embeddings. While all metrics contribute to a holistic assessment, we emphasize that pose fidelity and semantic alignment are the primary objectives of our task, as they directly reflect structural correctness and conditional control.

### 4.2 COMPARISON WITH PREVIOUS WORKS

**Quantitative and Qualitative Results.** Table 1 summarizes the quantitative comparison on the Human-Art dataset across six metrics. Specifically, KID values are multiplied by 100 for clarity. All baseline methods are trained for 10 epochs on Human-Art, except GRPose, which uses 20 epochs. To ensure fairness, we also report the 20-epoch version of SoftPose in the last two rows for direct comparison. As shown in the table, our method outperforms all baselines on pose accuracy (AP and CAP), with AP improving from 48.88 to 52.17 and CAP reaching 72.69. Compared to GRPose, SoftPose further improves AP from 49.50 to 56.31 and CAP from 70.84 to 73.19. While maintaining strong pose alignment, our method achieves competitive image quality with FID and KID values comparable to prior works These results demonstrate that our method achieves strong performance in pose accuracy, while maintaining competitive image fidelity and text alignment.

Figure 4 presents qualitative comparisons with several state-of-the-art methods on the above datasets. Our method consistently preserves fine-grained pose details in occluded regions (e.g., second and fourth rows), accurately maintains subject count and localization in crowded scenes (e.g., first row). Additionally, our approach recovers individual pose structures with greater precision, such as correctly distinguishing body orientation and subtle limb angles (e.g., third and fourth rows). Figure 5(b) further demonstrates the strong generalization capability of our method under varying subject counts and diverse pose scales. Even when the conditioning skeletons occupy very small areas in the image or appear in dense, crowded scenarios, our model can still accurately local-

Table 2: Results on datasets MHP and MPII. Comparison with two recent state-of-the-art methods focusing on pose accuracy under the same training setup.

| Dataset | Methods | AP (%) ↑ | CAP (%) ↑ | PCE ↓ | FID ↓ | KID ↓ | CLIP-score |
|---------|---------|----------|-----------|-------|-------|-------|------------|
| MHP-v2 | ControlNet Zhang et al. (2023b) | 58.17 | 67.55 | 1.32 | **3.11** | **0.37** | 31.65 |
| | StablPose Wang et al. (2024) | 71.83 | 69.26 | **1.18** | 3.27 | 0.46 | 31.61 |
| | GRPose Yin et al. (2025) | 65.54 | 68.74 | 1.21 | 3.34 | 0.49 | 31.53 |
| | SoftPose(Ours) | **72.35** | **73.36** | 1.21 | 3.45 | 0.51 | **31.71** |
| MPII | ControlNet | 41.33 | 70.51 | 0.82 | **4.57** | **1.25** | 31.47 |
| | StablePose Wang et al. (2024) | 53.70 | 75.18 | 0.77 | 4.61 | 1.32 | **31.48** |
| | GRPose Yin et al. (2025) | 43.74 | 71.31 | **0.73** | 5.51 | 1.50 | 31.27 |
| | SoftPose (Ours) | **54.42** | **78.27** | 0.77 | 5.25 | 1.40 | 31.36 |

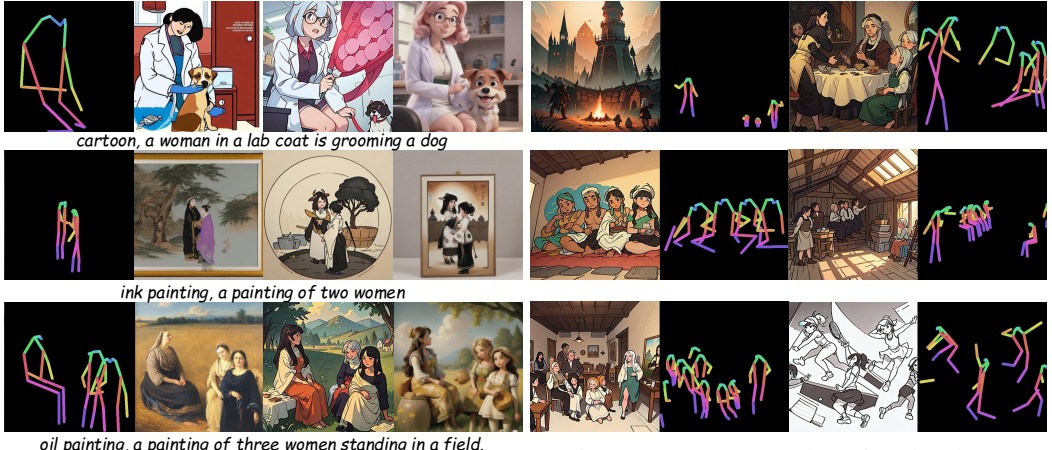

*cartoon, a woman in a lab coat is grooming a dog*

*ink painting, a painting of two women*

*oil painting, a painting of three women standing in a field.*

**Pose**     **SD 1.5**     **Anime Art**     **Disney**     **Generalization Across Varying Numbers of People and Pose Sizes**

Figure 5: (a)(Left) Qualitative Results Demonstrating the Compatibility of SoftPose with Various Base Models. (b)(Right) Demonstration of our method's generalization.

ize each person and preserve their anatomical structure. Furthermore, as shown in Figure 5(a), our approach demonstrates its flexibility with various base diffusion architectures.

**Multi-Human Adaptability and Single-Human Retention.** Table 2 presents quantitative results on the MHP v2.0 and MPII datasets, evaluating our method against two recent state-of-the-art baselines: StablePose and GRPose. The MHP dataset exclusively consists of images containing at least two people, making it particularly suitable for testing multi-person generation under complex spatial layouts and occlusion conditions. For a fair comparison, all methods were trained on MHP for 10 epochs. As shown in the table, our method achieves the highest AP (72.35%) and CAP (73.36%) among all competitors, demonstrating superior pose fidelity and inter-person consistency, with a comparable people count error (PCE). For the MPII dataset, which includes both single-person and multi-person scenes, the results highlight the robustness and generalization of our method across different subject counts. Our approach again attains the best AP (54.42%) and CAP (78.27%). These results confirm that SoftPose not only excels in crowded, multi-person scenarios but also retains strong performance in mixed single and multi-person cases, underscoring its adaptability and transferability across diverse datasets and scene complexities. Remarkably, inference time remains nearly constant across varying subject counts, thanks to our patch-level attention design. By operating on a fixed-resolution grid, computational cost is decoupled from the number of subjects, enabling efficient generation even in crowded scenes.

## 4.3 ABLATION STUDY

This section presents an ablation study evaluating the necessity of two core components: the Interaction-Aware Soft Attention Module (IA-SAM) and the Progressive Feature Injection Strategy (PFIS). As shown in Table 3, removing the IA-SAM leads to a significant performance decline, with AP dropping from 46.83% to 39.52% and CAP from 71.87% to 69.19%, underscoring its crucial role in differentiating individual skeletons and accurately transferring pose information in

Table 3: Ablation of each module in framework.

| Components | AP (%)↑ | CAP (%)↑ | PCE↓ |
|---|---|---|---|
| ControlNet | 39.52 | 69.19 | 1.54 |
| +IA-SAM | 46.83 | 71.87 | 1.55 |
| +IA-SAM+$\mathcal{L}_{IA}$ | 49.86 | **73.18** | 1.49 |
| +IA-SAM+$\mathcal{L}_{IA}$+PFIS | **52.17** | 72.69 | **1.47** |

Table 4: Ablation of kernel size.

| Kernel Size | AP(%)↑ | CAP(%)↑ | PCE↓ |
|---|---|---|---|
| (17, 11) | 49.17 | 72.28 | 1.51 |
| (21, 13) | 49.91 | 72.58 | 1.49 |
| (23, 13) | **52.17** | **72.69** | 1.47 |
| (23, 17) | 50.13 | 72.51 | **1.45** |

multi-person scenes. Qualitative results in Figure 6 confirm that without IA-SAM, the model confuses body orientation and merges overlapping limbs, producing anatomically implausible outputs. Building upon IA-SAM, the interaction-aware loss $\mathcal{L}_{IA}$ further enhances supervision by explicitly weighting interaction-centric regions during training. This results in a measurable gain in AP and achieves peak CAP, confirming its role in reinforcing spatial consistency where subjects interact.

The integration of PFIS further enhances structural fidelity by aligning coarse-to-fine pose conditioning with the diffusion process's generative trajectory. This staged refinement improves AP by 2.31 points, the highest among all variants, demonstrating its critical role in resolving fine-grained ambiguities such as occluded limbs and joint articulation. The marginal 0.49 point decrease in CAP reflects a deliberate design trade-off. PFIS prioritizes anatomical precision over global pose vector similarity, yielding more realistic interactions at the cost of minor cosine-distance degradation. As visualized in Figure 6, the trade-off manifests as sharper limb boundaries and more plausible contact geometries, qualities not captured by CAP but essential for perceptual realism. Together, IA-SAM and PFIS form an interdependent framework that enables accurate and coherent generation in densely interacting multi-person scenarios.

We also perform an ablation study on the kernel size $k$ within PFIS, fixing $\sigma = 3$ to ensure consistent smoothing strength across scales. By varying $k$, we modulate the spatial extent of pose masks: larger $k_1$ expands masks for global structure, while smaller $k_2$ retains local detail. This decoupling ensures that differences in scale arise solely from spatial coverage, not from variations in blur intensity. As demonstrated in Table 4, the configuration with $k_1 = 23$ for early steps and $k_2 = 13$ for late steps achieve optimal balance between layout coherence and interaction fidelity.

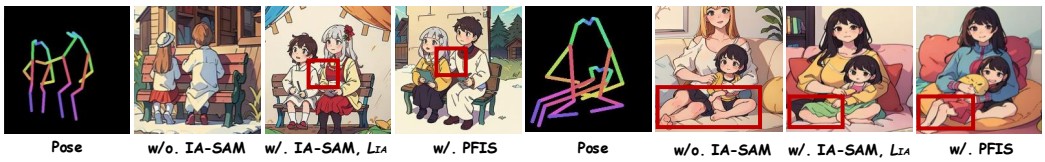

Figure 6: Two qualitative examples demonstrating the results of ablation study.

## 5 CONCLUSION

We present SoftPose, a flexible and robust framework for pose-guided multi-person image generation that effectively addresses the challenges of modeling inter-person spatial dependencies and handling complex interaction scenarios. SoftPose introduces an Interaction-Aware Soft Attention Module to emphasize interaction regions across multiple skeletons and explicitly capture spatial relationships among subjects. Additionally, a Progressive Feature Injection Strategy is employed to refine pose fidelity in a coarse-to-fine manner, enhancing structural details in multi-person scenes. Extensive experiments on diverse datasets demonstrate that SoftPose generalizes well across varying subject counts and spatial layouts, achieving consistent pose alignment and high-quality generation.

Despite its strengths, SoftPose inherits limitations from its Stable Diffusion backbone. It occasionally produces artifacts in hands and faces, which may be mitigated by adding facial and hand keypoints. And it may also misinterprets poses under rare viewpoints or with incomplete skeletons. While SoftPose achieves strong pose accuracy and text alignment, its FID and KID are slightly higher than some baselines. This difference can be partially attributed to known limitations of these metrics, particularly FID's reliance on Inception-v3 features, which are not well suited to evaluating condition-aligned generation and may overlook perceptual fidelity in structured content such as human poses. Notably, this gap is often imperceptible in visual inspection, as our generations maintain high structural coherence and interaction plausibility despite the metric discrepancy. Improving robustness in these scenarios remains a key direction for future work.

**Reproducibility Statement.** This work prioritizes reproducibility, and we have made every effort to ensure our findings can be independently verified. Our implementation details, including model architecture, training procedures, and hyperparameter settings, are thoroughly documented in the main text and Appendix. We will provide our complete source code, including code for data preprocessing and model training, as an anonymous downloadable file in the supplementary materials to facilitate replication. Additionally, a detailed description of our evaluation protocol and all metrics used can be found in A.1 and 4.1 to enable consistent performance assessment.

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

## A APPENDIX

This appendix provides additional details and results that could not be included in the main paper due to page limitation. It includes the following sections:

- Experimental details in Sec. A.1.
- Additional quantitative results in Sec. A.2.
- Additional qualitative results in Sec. A.3.
- Future work and discussion in Sec. A.5.
- Statement of the use of LLMs in Sec. A.6.

### A.1 EXPERIMENTAL DETAILS

#### A.1.1 DATASET AND PREPROCESSING DETAILS

We conduct experiments on three representative datasets: Human-Art Ju et al. (2023a), MHP v2.0 Zhao et al. (2020), and MPII Andriluka et al. (2014). All datasets are preprocessed to follow the unified annotation format used in HumanSD Ju et al. (2023b), which includes 2D keypoints and text descriptions.

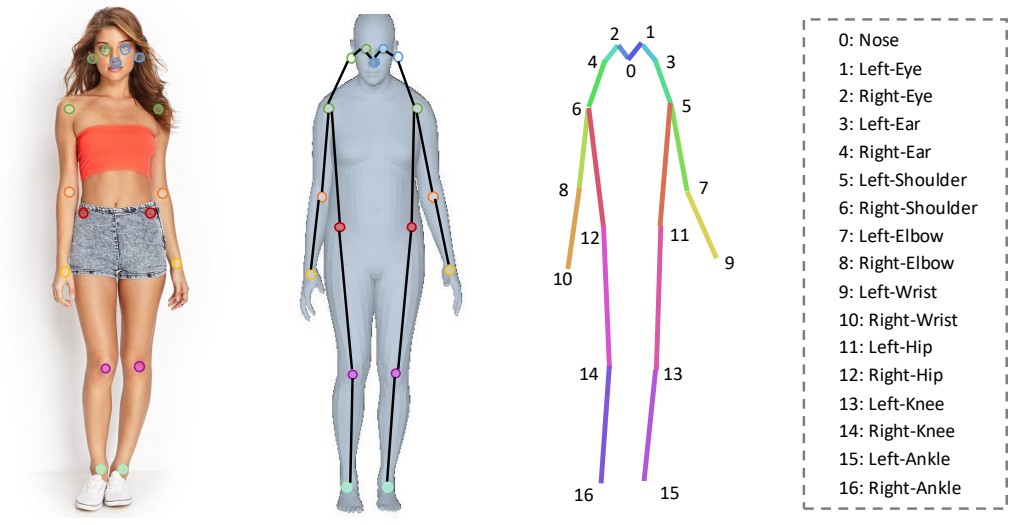

Figure 7: Visualization of the extracted keypoints and their associated information.

**Human-Art.** Human-Art contains over 50K high-quality images with 123K+ human instances from both natural and artistic domains (5 real-world and 15 synthetic scenarios). Each human instance is annotated with 21 2D keypoints, self-contact indicators, bounding boxes, and descriptive captions. To ensure consistency across datasets, we follow the format defined in HumanSD Ju et al. (2023b) and retain only 17 keypoints per person. The retained keypoints and their corresponding information are illustrated in Figure 7.

**MHP-v2.** The MHP-v2 dataset comprises 25,403 images, each containing at least two people (average of 3), with a total of 40K+ human instances. However, the original keypoint annotations in MHP lack facial landmarks such as the nose, eyes, and ears. To address this, we use DWpose Yang et al. (2023) to supplement facial keypoints and convert all annotations into the HumanSD-compatible format. We further perform manual filtering to remove samples with poor facial detection quality. We finally have 15,300 images for training and 4,999 for testing. Moreover, since MHP v2.0 does not include textual descriptions, we leverage Qwen-VL, a vision-language model, to generate natural language captions that match the Human-Art style. These descriptions provide contextual cues for each image, ensuring consistency in multimodal supervision across datasets.

**MPII.** The MPII dataset contains around 25K images depicting over 40K people across 410 everyday activities. Each image is extracted from a YouTube video and provided with a short temporal context. Since MPII's pose annotations also lack facial keypoints, we apply the same preprocessing pipeline as used for MHP, i.e., DWpose enhancement and manual filtering. We randomly select 4,115 multi-person images as our test set. To ensure consistency and fairness in data representation, we follow the same procedure as with MHP by using Qwen-VL to generate textual descriptions that match the Human-Art annotation style.

### A.1.2 EVALUATION METRICS

To provide a clearer understanding of our evaluation protocol, we offer detailed explanations of metrics used in our experiments below, highlighting their purpose and what aspect of model performance they capture.

**Pose Accuracy.** To evaluate pose accuracy in generated images, we adopt the COCO Lin et al. (2014) evaluation protocol, which relies on the Object Keypoint Similarity (OKS). OKS measures the normalized distance between generated keypoints and ground truth, defined as:

$$\text{OKS} = \frac{\sum_i \left[ \exp\left(-\frac{d_i^2}{2s^2\kappa_i^2}\right) \cdot \delta(v_i > 0) \right]}{\sum_i \left[\delta(v_i > 0)\right]}, \tag{6}$$

where $d_i$ denotes the Euclidean distance between corresponding keypoints, $s$ is the object scale, $\kappa_i$ controls the falloff, and $v_i$ is the visibility flag of the ground truth keypoint. Based on OKS, we compute Average Precision (AP) to quantify pose alignment between generated images and ground truth. The keypoints used in this evaluation are detected using mmpose. However, AP remains sensitive to absolute spatial location. To complement this, we introduce CAP, which instead focuses on the angular similarity of poses, capturing whether the generated human articulation matches the target action regardless of spatial offset. Benefiting from our interaction-aware soft attention mechanism, our method achieves higher pose accuracy scores than prior work, highlighting better preservation of fine-grained pose details in complex multi-person scenarios.

**Text Image Consistency.** To quantitatively assess the semantic alignment between generated images and their corresponding text prompts, we employ the pre-trained clip-vit-large-patch14 Radford et al. (2021) model to extract feature embeddings from both modalities. Specifically, image features and text features are encoded into a shared latent space, allowing for a direct similarity comparison. This method provides a robust metric for assessing how well the visual content generated by our model aligns with the input textual descriptions, thereby serving as an effective measure of text-image consistency.

### A.1.3 DETAILS OF TRAINING AND EVALUATION.

Our experiments are conducted on four NVIDIA RTX 4090 GPUs, where training SoftPose on the HumanArt dataset with 33,250 samples for 10 epochs takes approximately 25 hours. The average peak GPU memory usage during training is around 20,023 MiB. For inference, our method generates each image in roughly 8.19 seconds on single 3090 GPU.

### A.2 ADDITIONAL QUANTITATIVE RESULTS

In this section, we present additional ablation study to further analyze the effectiveness of key components in our method. All experiments are conducted by training our model on the HumanArt dataset for 10 epochs, following the same training pipeline described in the main paper. These results provide deeper insights into how different design choices impact the final performance.

**Ablation on the Number of Training Stages.** We study the effect of varying the number of training stages, as reported in Table 5. In our method, the training process is divided into stages according to the diffusion timestep, allowing different scales of information to be injected at different temporal segments. Specifically, we compare three settings: (1) using a single stage where the same guidance is applied uniformly across all $T$ timesteps; (2) dividing the process into two stages, with separate conditioning for the first $T//2$ and the last $T//2$ timesteps; and (3) splitting into three stages ($T//3$

for a finer granularity of control. Increasing the number of stages generally improves performance by enabling more progressive and stage-aware refinement. However, excessive splitting may introduce redundant computation or risk overfitting, offering limited gains beyond a certain point. Our default choice achieves a balanced trade-off between efficiency and pose alignment quality, leading to stable and high-quality results.

Table 5: Performance Comparison Across Different Numbers of Training Stages in PFIS.

| Stages | AP ↑ | CAP ↑ | PCE ↓ | CLIP-score ↑ |
|---|---|---|---|---|
| 1 ($T$) | 49.86 | **73.18** | 1.49 | **32.58** |
| 2 ($T_1, T_2$) | **52.17** | 72.69 | **1.47** | 32.50 |
| 3 ($T_1, T_2, T_3$) | 50.49 | 71.90 | 1.55 | 32.57 |

## A.3 ADDITIONAL QUALITATIVE RESULTS

In this section, we provide further qualitative results to complement the quantitative analysis presented in the main paper. These results illustrate the robustness and versatility of our method across both single-person and multi-person scenarios. By showcasing a variety of challenging conditions, we demonstrate the model's ability to generate realistic, coherent poses while maintaining high visual quality, even in the presence of occlusions, rare viewpoints, or complex interactions between multiple subjects.

**Additional Qualitative Results on Single-Person Scenarios.** Building on the results presented in the main paper, we further demonstrate that our method not only excels in multi-person scenarios but also retains strong performance in single-person scenarios. As shown in Figure 8, our approach effectively captures fine pose details, even under challenging angles or subtle body details, while maintaining high visual quality across various settings.

**Additional Qualitative Results on Multi-Person Scenarios.** We present additional qualitative results on multi-person scenarios in Figure 9 to further highlight the robustness and adaptability of our method in capturing complex interactions between multiple subjects. These examples demonstrate the model's ability to accurately generate realistic and coherent poses for each individual, even in the presence of overlapping bodies, occluded joints, or intricate poses, ensuring seamless integration and consistency across all subjects in the scene.

**Additional Qualitative Results on MHP dataset.** We provide additional qualitative comparisons on the MHP-v2 dataset, which contains real-world multi-person scenes with diverse interactions. As shown in Figure 10, our method maintains quite good quality under realistic imaging conditions, further validating its effectiveness beyond artistic domains.

## A.4 USER STUDY

We conduct an extensive user study to compare our method with three baselines: ControlNet Zhang et al. (2023b), StablePose Wang et al. (2024) and GRPose Yin et al. (2025). We use 100 images from the validation set, 12 participants are asked to compare each pair of methods based on four criteria: pose fidelity, interaction quality (multi-person only), overall visual quality, and overall preference. As shown in Table 6, our method significantly outperforms the previous methods. Specifically, users' feedback reveals that despite achieving lower FID scores, our method is rated as essentially equivalent in overall visual quality, demonstrating that the FID gap is imperceptible to human observers.

We conduct our user study in a questionnaire format to collect user preferences for different methods. We observe that in pose-guided generation tasks, users frequently encounter cases where both methods achieve comparable pose accuracy, particularly in simple single-person scenarios. Therefore, we allowed participants to select "equally good" for pose fidelity and interaction quality metrics. Additionally, Human-Art contains several challenging cases with heavy occlusion or complex interactions where all methods fail to generate satisfactory results. Consequently, we allow users to select

"neither is good" for overall quality and overall preference metrics. We implement a single-blind mechanism where the corresponding method for each question is randomly sampled, and the left-right positions of the two results are randomized, ensuring fairness in the comparison. We collect over 600 comparison results (300 comparison instances × 2 raters) and calculate our method's win rate after excluding cases where both methods are rated equally good or neither is good.

Table 6: User study results comparing our method with three baselines. Win rates are calculated after excluding "equally good" and "neither is good" responses. Interaction Quality is evaluated only on 70 multi-person samples while there are 30 single-person samples.

| Ours vs. | Pose Fidelity ↑ | Interaction Quality ↑ | Image Quality ↑ | Overall Preference ↑ |
|---|---|---|---|---|
| ControlNet | 71.8% | 75.3% | 49.2% | 66.4% |
| StablePose | 62.5% | 65.1% | 50.6% | 59.7% |
| GRPose | 65.8% | 68.4% | 50.1% | 62.3% |

## A.5 FUTURE WORK AND DISCUSSION

**Failure Cases.**  Despite the method's effectiveness, failure cases arise when handling rare or incomplete poses. For example, as shown in Figure 11 , the head angle is inaccurately generated due to unusual pose angles not covered in training. Similarly, missing skeletal information sometimes lead to unrealistic pose generation. These issues stem from the model's reliance on complete skeletons and its limited generalization to extreme or incomplete poses. While these cases show limitations, they do not undermine the overall effectiveness of our approach in handling typical poses.

**Future Work.**  While SoftPose demonstrates strong performance in most scenarios, several limitations warrant further exploration. One key avenue for improvement is the transfer of our method to a base model that is specifically optimized for fine-grained details such as hands and faces. By leveraging a more specialized backbone, we can enhance the accuracy and realism in these areas. Additionally, enriching the pose representation with more detailed facial and hand keypoints could help mitigate current artifacts and improve the model's performance in fine-grained regions. Another challenge lies in handling rare viewpoints and incomplete skeletal annotations, particularly in multi-person settings where occlusions and pose distortions are more prevalent. To address these issues, we propose increasing the sampling ratio of data with uncommon viewpoints to alleviate the problem of rare viewpoints, while leveraging data augmentation to enhance the model's generalization capability for incomplete skeletal annotations. Future work will focus on developing better methods to improve generation quality in these extreme scenarios, enhancing the model's robustness and ensuring more accurate results in challenging conditions. Moreover, though our method is designed to be architecture-agnostic, operating on universal attention mechanisms. We acknowledge that comprehensive validation on state-of-the-art models (SDXL, Flux, SD3, Qwen-Image) is important future work. While our current evaluation focuses on SD1.5 for fair comparison with established baselines, extending to DiT-based architectures and higher-resolution models will more fully demonstrate the generalizability of our approach across diverse backbones.

## A.6 STATEMENT

**The Use of LLMs.**  We acknowledge the use of a large language models (LLMs) as a general-purpose writing assistant during the preparation of this manuscript. The LLMs were used exclusively for refining and improving the clarity, grammar, and style of the text.

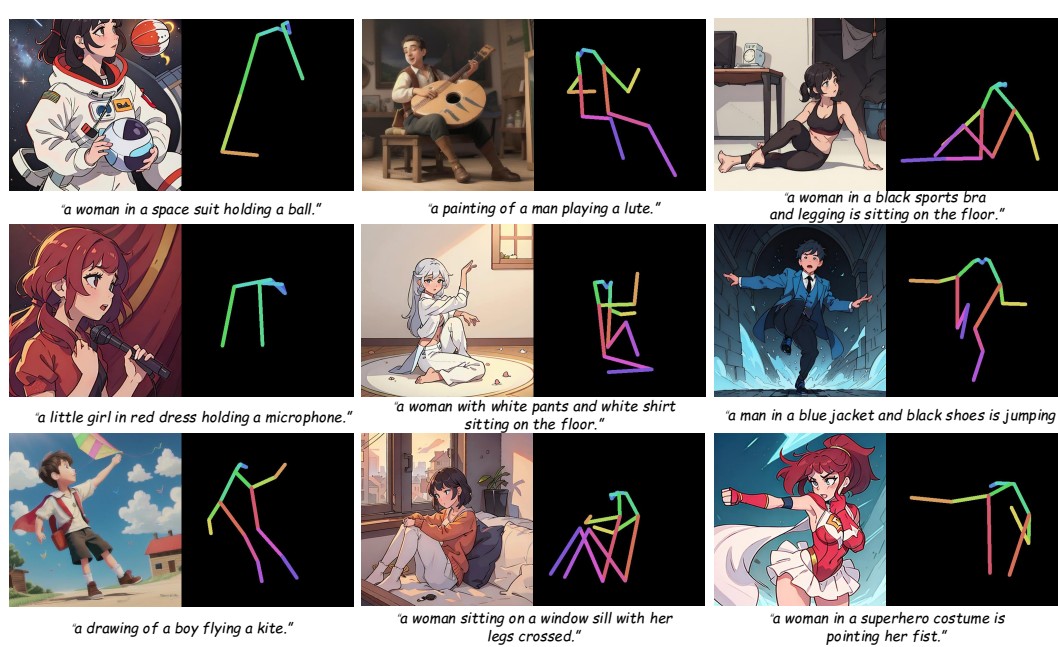

Figure 8: Additional qualitative results on single-person scenarios.

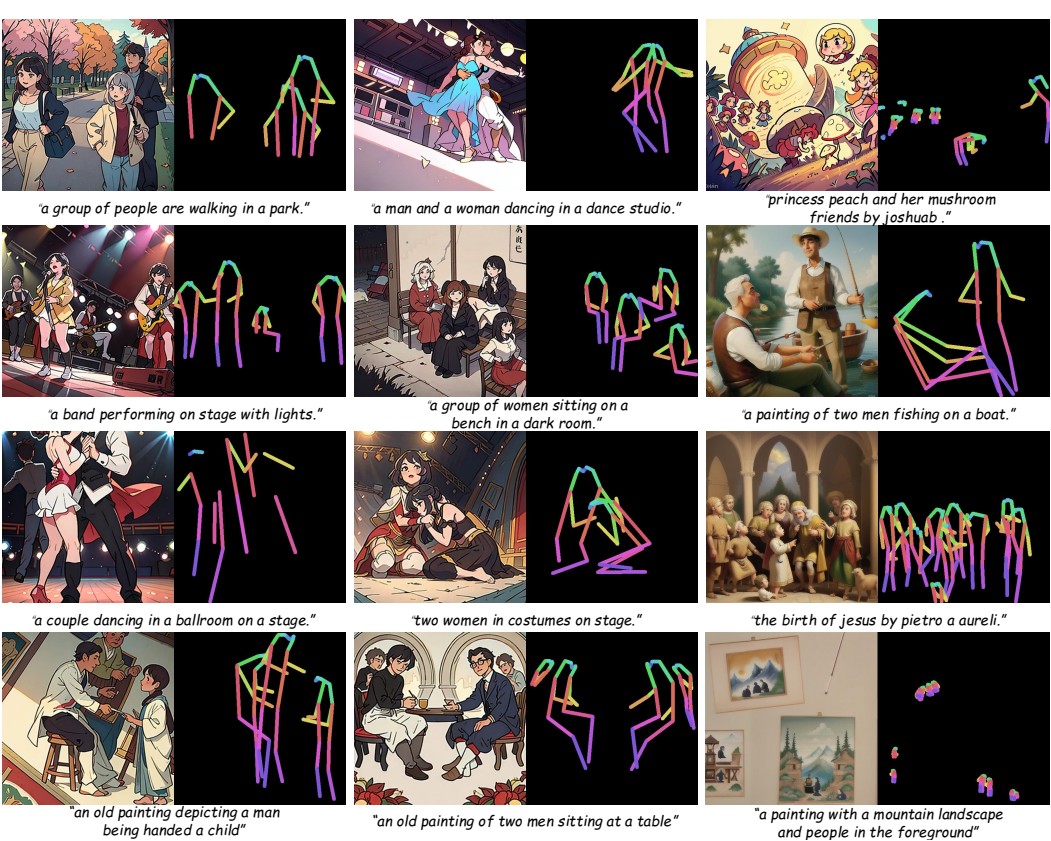

Figure 9: Additional qualitative results on multi-person scenarios.

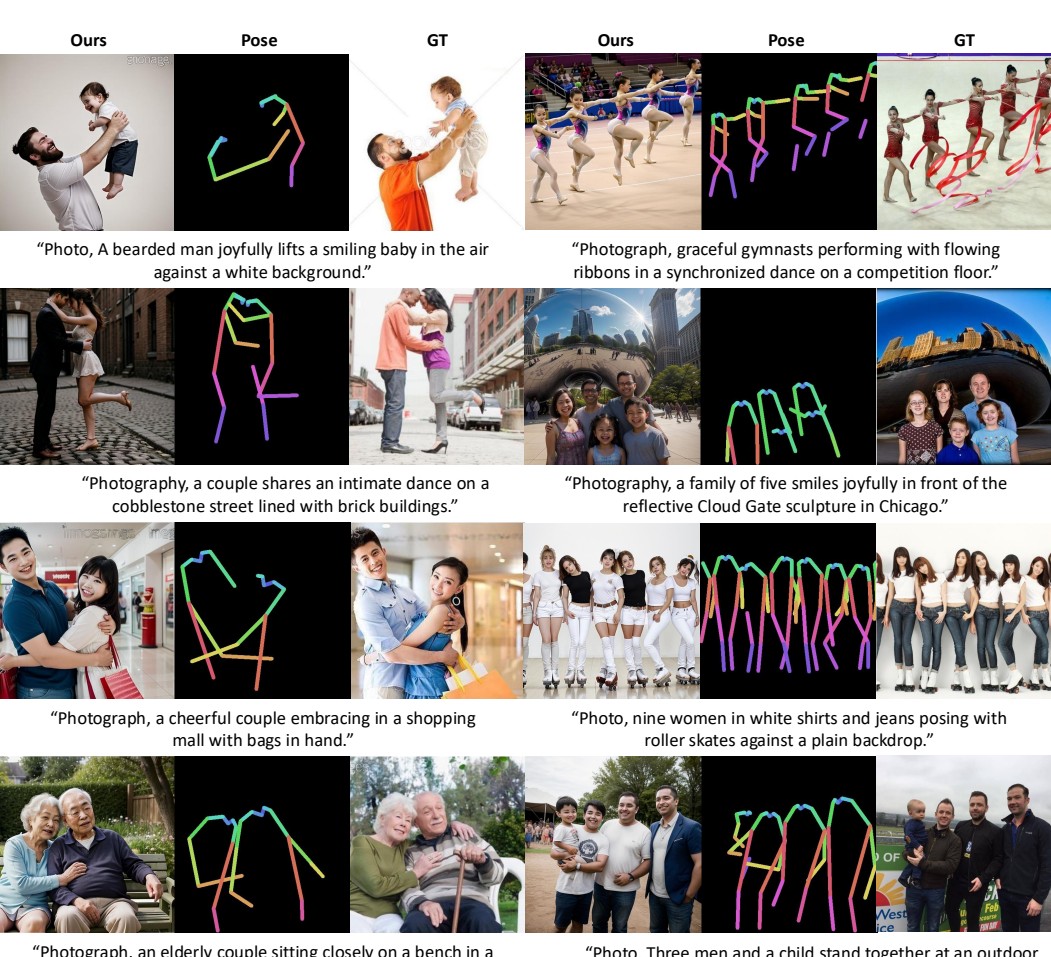

Figure 10: Additional qualitative results on MHP dataset.

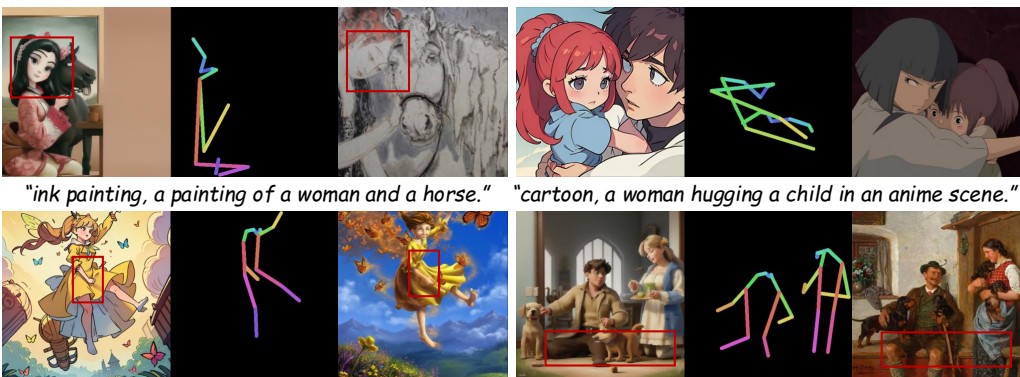

"ink painting, a painting of a woman and a horse."    "cartoon, a woman hugging a child in an anime scene."

"digital art, a girl in a yellow dress is flying through the air with butterflies"    "oil painting, a painting of a man and woman with two dogs"

Figure 11: Failure case examples.

