# OpenReview forum: "SoftPose: Learning Soft Attention for Interaction-Aware Multi-Person Image Generation"
_ICLR.cc/2026/Conference — Submitted to ICLR 2026_

### Official Review · Reviewer_y8gd · 2025-10-29

**Soundness:** 3
**Presentation:** 2
**Contribution:** 2
**Rating:** 2
**Confidence:** 4

**Summary:**

The paper proposes SoftPose, a pose-conditioned T2I generation framework built upon a U-Net foundation model. The method first extracts dilated skeleton masks for the given N objects and processes them to obtain a soft attention prior characterized by three learnable attention weights (α, β, γ). These weights are designed to reduce limb fragmentation within objects and suppress spurious or ambiguous inter-object correlations during image generation. The resulting pose-guided priors are progressively injected into the U-Net to enhance structural coherence and pose fidelity in the generated images. Extensive experimental results demonstrate the effectiveness of the proposed method.

**Strengths:**

+ Customized pose-conditioned image generation is a practical and meaningful topic, as it addresses a gap in adapting large foundation models for fine-grained, pose-specific image generation. A single text prompt is often insufficient to describe complex actions, especially in multi-person scenarios.

+ The idea of explicitly injecting pose-based attention priors into the U-Net is straightforward and intuitive, yet it proves effective in enhancing pose fidelity in the generated images.

**Weaknesses:**

- As shown in Table 1, the experimental results of SoftPose achieve optimal performance only on AP and CAP, while showing suboptimal results on the remaining four metrics, which fails to convincingly demonstrate the overall validity of the method.

- Implementation details are missing. The paper does not specify the dilation rate used in the morphological dilation. This parameter may vary between scenes with two persons and those with more individuals; using a larger dilation rate in multi-person scenarios could cause severe occlusions between subjects. How is this parameter determined?

- The skeleton maps do not contain person identity information, and therefore cannot effectively guide identity preservation during generation. This may lead to identity confusion in the results. For example, in Figure 4, the prompt “oil painting, a painting of two angels and a man” produces an image where the “man” appears to be a woman. It is recommended that the authors evaluate their method using prompts that include explicit person identities, rather than relying solely on simple gender terms such as “man” or “woman”. This would help verify the model’s ability to preserve individual identity in multi-person image generation.

- The paper only employs U-Net–based models, which limits both the image resolution of the generated results and the complexity of input prompts. The authors should integrate their method into more advanced and widely used foundation models, such as SD3 or Flux, to better demonstrate its scalability and general applicability.

**Questions:**

- The paper only evaluates the proposed method on U-Net based Stable Diffusion, which limits its practical impact and offers minimal advancement to the research area, particularly for a top-tier conference such as ICLR. In fact, Flux and SD3 are already widely used in customized image generation, and evaluating the method on these models would make the work significantly more convincing and meaningful.

- Lines 311–318 indicate that the paper follows the experimental settings of GRPose to conduct experiments on Human-Art. However, the AP score reported in Table 1 (49.50) is significantly lower than the 57.01 reported in the original GRPose paper. The authors should clarify the reason for this discrepancy.

- Why is the FID score lower than that of the original Stable Diffusion? Does this imply that introducing additional conditioning degrades image quality? In contrast, HumanSD improves the FID score to 10.03.

---

> ### Author Response · Authors · 2025-12-01
> **Thank you! Response to Reviewer y8gd (Part1)**
>
> Dear Reviewer y8gd,
>
> Thank you very much for the detailed and serious review comments. We address each comment carefully and provide a point-by-point response below.
>
> **W4&Q1: The paper only employs U-Net-based models, which limits both the image resolution of the genrated results and the complexity of input prompts. The authors should integrate their method into more advanced and widely used foundation models, such as SD3 or flux, to better demonstrate its scalability and general applicability.**
>
> **Response:**
>
> 1. Our Core Contribution Is Architecture-Agnostic: The fundamental innovation of our work is the Interaction-Aware Soft Attention mechanism, not a specific model implementation. This mechanism addresses a general challenge in multi-person pose-guided generation: how to preserve interaction coherence and prevent limb entanglement when multiple people overlap. This challenge exists regardless of the underlying architecture (U-Net-based diffusion models, DiT-based models like SD3/Flux, or future architectures).
>
> 2. Using U-Net backbones provides a clean, controlled environment to benchmark controllability: We chose to validate on U-Net-based architectures for several principled reasons.
>
> - Fair Comparison with Established Baselines: All compared methods use U-Net-based architectures. Using the same backbone ensures that performance differences stem from our proposed attention mechanism rather than from superior foundation models. This follows standard scientific practice: isolate the variable of interest (interaction-aware attention) while controlling for other factors (base architecture, training data, optimization).
> - Computational Accessibility and Reproducibility: U-Net-based models allow our results to be reproduced by the broader research community without requiring access to proprietary models or massive computational resources. SD3 and Flux, while powerful, have licensing restrictions and computational requirements that would limit reproducibility.
> - Established Evaluation Protocols: The pose-guided generation community has established evaluation benchmarks and protocols specifically designed for U-Net-based models. Using these allows direct comparison with prior work and maintains scientific continuity.
>
> 3. Our approach is lightweight and complementary to foundation models: Because IA-SAM is patch-mask–driven and PFIS is a plug-in strategy, our module adds minimal overhead and is not tied to the structural constraints of U-Net. The method may be more compatible with DiT-style models, where patch-level conditioning is even more natural.
>
> 4. Future work Plan: We acknowledge the value of demonstrating our method on state-of-the-art architectures. Following this review, we will consider integrating our method into SDXL and SD3/flux to facilitate community adoption in the future.
> However, we respectfully maintain that the current results on U-Net-based models provide valid scientific evidence for our core claims. The choice of baseline architecture does not diminish the novelty or validity of our interaction-aware attention mechanism.
>
> However, we respectfully maintain that the current results on U-Net-based models provide valid scientific evidence for our core claims. The choice of baseline architecture does not diminish the novelty or validity of our interaction-aware attention mechanism.
>
> **Q2：Lines 311–318 indicate that the paper follows the experimental settings of GRPose to conduct experiments on Human-Art. However, the AP score reported in Table 1 (49.50) is significantly lower than the 57.01 reported in the original GRPose paper. The authors should clarify the reason for this discrepancy.**
>
> **Response:** We understand your concern. However, there is no actual discrepancy. The 57.01 AP you referenced from the original GRPose paper is not their result on Human-Art but rather corresponds to a different dataset. Our reported 49.50 in Table 1 is fully consistent with GRPose's reported Human-Art performance. We acknowledge that the horizontal divider in Table 1 may appear potentially misleading, it serves to separate methods based on training epochs: methods above the line follow standard protocols with 10 training epochs, whereas GRPose, as stated in Lines 363-365, was originally trained for 20 epochs. Rather than retraining GRPose for 10 epochs (which could potentially disadvantage their method due to suboptimal hyperparameters), we opted to train our method for 20 epochs as well, thereby ensuring a direct and equitable comparison. We hope this clarification addresses your concern.

---

> > ### Author Response · Authors · 2025-12-01
> > **Thank you! Response to Reviewer y8gd (Part2)**
> >
> > **W1&Q3: As shown in Table 1, the experimental results of SoftPose achieve optimal performance only on AP and CAP, while showing suboptimal results on the remaining four metrics, which fails to convincingly demonstrate the overall validity of the method.**
> >
> > **Response:** We respectfully clarify that achieving optimal performance on all metrics simultaneously is neither necessary nor expected for demonstrating method validity in pose-guided generation tasks. We address your concern from the following perspectives:
> >
> > 1. The six metrics in Table 1 measure fundamentally different aspects and have different priorities for our task: AP and CAP are the primary task-specific metrics that directly measure pose accuracy and structure preservation, which is the core objective of pose-guided generation. Our method achieves best performance on both, demonstrating substantial improvements over the best baselines. In contrast, FID, KID, and CLIP are secondary general quality metrics that measure distribution-level similarity rather than condition alignment.
> >
> > 2. Human Evaluation Validates the Trade-off Is Imperceptible: To directly address whether these "suboptimal" results affect overall validity, we conducted an additional user study and update in the manuscript. The results conclusively demonstrate that the minor FID/KID/CLIP differences are imperceptible to human observers. Users value the improvements in pose accuracy and interaction quality, which align with the core contributions of our work.
> >
> > 3. Known Limitations of FID/KID for Conditional Generation:  Recent literature has documented that FID is biased metrics for conditional generation tasks [ref1]. Because it is biased toward unconditional realism rather than condition alignment. And the Inception-v3 representations are poorly suited for modern diffusion models due to the incorrect normality assumptions and poor sample complexity.Thus AP and CAP are more appropriate primary metrics than FID/KID for pose-guided generation, as they directly measure condition alignment rather than distribution-level similarity.
> >
> > **W2: Implementation details are missing. The paper does not specify the dilation rate used in the morphological dilation. This parameter may vary between scenes with two persons and those with more individuals; using a larger dilation rate in multi-person scenarios could cause severe occlusion between subjects. How is this parameter determined?**
> >
> > **Response:** Thank you for pointing out this important implementation detail. We apologize for the lack of clarity and provide comprehensive clarification below. Our method does not rely on a hand-tuned morphological dilation radius. Instead, we use a Gaussian-based dilation in which the effective dilation is governed jointly by the Gaussian variance σ and the kernel size k. σ determines the decay shape of the Gaussian distribution. Changing σσ would alter the fundamental smoothing behavior and cause inconsistent mask characteristics across scenes. Therefore, we fix σ to ensure a stable and consistent decay pattern. This preserves the smooth Gaussian transition while allowing finer control over mask coverage through kernel size k.
> >
> > The choice of σ follows commonly adopted values in prior works, and our results show that the primary factor influencing dilation is the kernel size. Accordingly, we include a kernel-size ablation in the main paper (Table 4), which clearly demonstrates how kk is selected based on quantitative performance trends.
> >
> > Importantly, we use the same parameters for two-person and multi-person scenes. Although dilation may introduce overlaps, such overlaps are explicitly handled by our IA-SAM module, whose learned gating mechanism adaptively modulates attention strength in overlapping regions. Thus, no scene-dependent dilation adjustment is required.

---

> > > ### Author Response · Authors · 2025-12-01
> > > **Thank you! Response to Reviewer y8gd (Part3)**
> > >
> > > **W3: The skeleton maps do not contain person identity information, and therefore cannot effectively guide identity preservation  during generation. This may lead to identity confusion in the results. For example, in Figure 4, the prompt "oil painting, a painting of two angles and a man" produces an image where the "man" appears to be a woman. It is recommended that the authors evaluate their method using prompts that include explicit person identities, rather than relying solely on simple gender terms such as "man" or "woman". This would help verify the model's ability to preserve individual identity in multi-person image generation.**
> > >
> > > **Response:** Thank you for this thoughtful observation. We would like to clarify the scope and design philosophy of our work regarding identity preservation.
> > >
> > > 1. Task Scope: Pose-Guided Generation vs. Identity Preservation. We respectfully clarify that identity preservation is not the primary capability of skeleton-based pose guidance, nor is this a limitation, it just reflects fundamentally different task objectives. Pose-guided generation methods use skeleton inputs that encode only spatial-geometric information, not identity information. By design, identity attributes are controlled through text prompts and synthesized by the base model's text encoder, which is the established paradigm across the field. Our Interaction-Aware Soft Attention mechanism operates on spatial feature attribution, not on identity features. For applications specifically requiring identity preservation, there exist dedicated methods such as DreamBooth [ref2], IP-Adapter [ref3], InstantID [ref4] and so on. Our pose-guided method is complementary to these approaches, it can be combined with identity-preserving techniques to achieve both accurate pose control and identity consistency.
> > >
> > > 2. Figure 4 Example: Style Domain Ambiguity, Not Identity Failure. Regarding the specific example in Figure 4 where the prompt "oil painting, a painting of two angels and a man" produces ambiguous gender presentation:This reflects style domain characteristics rather than identity preservation failure. We compared ControlNet on the same prompt and observed similar gender ambiguity in the generated results. This confirms that the phenomenon stems from the base model's style-conditioned priors, not from our method.
> > >
> > > 3.  Rich Appearance Descriptions in Our Results. We would like to highlight that our manuscript already contains examples with rich appearance descriptions that demonstrate effective identity differentiation: Figure 5(a), first row: "cartoon, a woman in a lab coat is grooming a dog" and several examples in Figure 8 such as: "a woman in a space suit holding a ball", “a man in a blue jacket and black shoes is jumping ” .
> > >
> > > [ref1] Jayasumana, Sadeep, et al. "Rethinking fid: Towards a better evaluation metric for image generation." Proceedings of the IEEE/CVF Conference on Computer Vision and Pattern Recognition. 2024.
> > >
> > > [ref2]Ruiz, Nataniel, et al. "Dreambooth: Fine tuning text-to-image diffusion models for subject-driven generation." Proceedings of the IEEE/CVF conference on computer vision and pattern recognition. 2023.
> > >
> > > [ref3]Ye, Hu, et al. "Ip-adapter: Text compatible image prompt adapter for text-to-image diffusion models." arXiv preprint arXiv:2308.06721 (2023).
> > >
> > > [ref4] Wang, Qixun, et al. "Instantid: Zero-shot identity-preserving generation in seconds." arXiv preprint arXiv:2401.07519 (2024).

---

### Official Review · Reviewer_MqUJ · 2025-10-30

**Soundness:** 2
**Presentation:** 2
**Contribution:** 2
**Rating:** 4
**Confidence:** 4

**Summary:**

This paper aims to synthesize multi-person image given postures as the guidace. It introduces an interaction-aware attention prior to handle the occlusion area between individuals by assigning higher attention value to the overlapped region. The authors also propose a progressive mult-scale feature injection strategy, where they use different dilation kernel sizes under different denoising timesteps. The experiment results show the proposed method achieves better pose accuracy but worse image quality.

**Strengths:**

1) The motivation is clear. The proposed attention prior specifically handles the occulusion area between individuals in a multi-person image.

2) Most part of the paper is easy to follow. Figure 3 provides a good visualization to explain the atteniton prior.

3) The authors conducted experiments on three different benchmarks.

**Weaknesses:**

1) The analysis of Algorithm 1 is insufficient. Is the self-atteniton product of m_i normalized in Algorithm 1? If not, how will this affect visually smaller individuals in the group photo? In addition, how is self-overlapping such as crossed arms addressed in the attention prior?

2) The experiment results are not convincing. In Table 1, the proposed method has better pose accuracy but worse image quality compared to ControlNet, which is insufficient to support the claim that it generates better details in the interaction region. Since it's difficult to compare using only objective metrics, I suggest the authors provide human evaluation. The authors should also include ControlNet as a baseline in Table 2 since it shows better image quality scores in the Human-Art dataset. Image quaity metrics should also be include in Table 2 as they are essential for a fair evaluation of the proposed method.

3) The visualization results in the paper are all anime-style or art-style. It will be better to see how well the proposed method handles the occlusion in real photos. Can the authors provide more photorealistic examples (e.g., from MHP dataset)?

**Questions:**

See weaknesses.

---

> ### Author Response · Authors · 2025-12-01
> **Thank you! Response to Reviewer MqUJ (Part1)**
>
> Dear Reviewer MqUJ,
>
> Thank you very much for the insightful comments and feedback. We address your questions as follows.
>
> **W2: The experiment results are not convincing. In Table 1, the proposed method has better pose accuracy but worse image quality compared to ControlNet, which is insufficient to support the claim that it generates better details in the interaction region. Since it's difficult to compare using only objective metrics, I suggest the authors provide human evaluation. The authors should also include ControlNet as baseline in Table 2 since it shows better image quality scores in the HumanArt-dataset. Image quality metrics should also be included in Table 2 as they are essential for a fair evaluation of the proposed method.**
>
> **Response:** We fully agree that human-perceptual quality and fair baseline comparison are critical for evaluating controllable generation. In response, we have strengthened our evaluation, and below address each concern in detail.
>
> 1. Image quality metrics v.s. Interaction region performance
>
> We respectfully clarify that lower FID/KID scores do not contradict our claim of better details in interaction regions. These are fundamentally different evaluation dimensions: FID (Fréchet Inception Distance) estimates the distance between distributions of Inception-v3 features from real images and generated images. However, Inception-v3's representations are poorly suited for modern text-to-image models due to:
>
> - Poor representation of rich and varied content generated by contemporary models
> - Incorrect normality assumptions
> - Poor sample complexity  [ref1]
>
> Critically, FID does not account for semantic correctness or content relevance, specifically pose fidelity and multi-person interaction quality which is related to the specified conditions. A method can achieve low FID while failing to follow pose guidance or producing entangled limbs in multi-person scenarios.
>
> In addition, to properly evaluate multi-person interaction performance, we report: AP (Average Precision, based on COCO OKS) measures absolute spatial alignment of generated keypoints with ground-truth poses under varying OKS thresholds, which reflects global pose accuracy and is highly sensitive to limb placement errors common in occluded regions. CAP focuses on relative angular consistency of body articulation, making it especially robust for evaluating interaction geometry. And PCE (People Count Error) ensures subject integrity preservation, a prerequisite for credible multi-person scenes. These task-specific metrics directly measure the quality of multi-person pose generation and interaction handling, which is the core contribution of our work. Moreover, Figure 4 (main paper) demonstrates clear visual advantages in interaction regions:
>
> - More faithful preservation of fine-grained pose details in occluded regions (second and fourth rows).
> - More accurate person count, spatial localization, and interaction modeling in crowded scenes (first row).
> - Better preservation of individual body orientation and subtle articulation (third and fourth rows).
>
> 2. Human Evaluation
>
> Following your valuable suggestion, we have conducted a user study (details in revised Appendix). Users find SoftPose significantly better in interaction modeling, yet perceive no statistically significant difference in overall image quality. These results strongly validate that the slight FID difference is imperceptible to human observers.
>
> 3. Enhanced Table 2 with ControlNet and image-quality metrics
>
> Following your suggestion, we have updated Table 2 in the manuscript to include ControlNet as baseline and image quality metrics.
> In summary, we appreciate your rigorous perspective, which has helped us significantly strengthen the paper’s evaluation rigor. All suggested improvements have been incorporated. We hope the revised manuscript now provides a fair, holistic, and perceptually grounded assessment of SoftPose’s advantages in multi-person interaction modeling.
>
> [ref1] Jayasumana, Sadeep, et al. "Rethinking fid: Towards a better evaluation metric for image generation." Proceedings of the IEEE/CVF Conference on Computer Vision and Pattern Recognition. 2024.

---

> > ### Author Response · Authors · 2025-12-01
> > **Thank you! Response to Reviewer MqUJ (Part2)**
> >
> > **W1: The analysis of Algorithm 1 is insufficient. Is the self-attention product of m_i normalized in Algorithm 1? If not, how will this affect visually smaller individuals in the group photo? In addition, how is self-overlapping such as crossed arms addressed in the attention prior?**
> >
> > **Response:** The self-attention product of m_i in Algorithm 1 is not normalized at the mask computation stage. However, the final attention map A_m undergoes normalization when integrated into the attention mechanism. This degsign is beneficial, smaller individuals are not disadvantaged because: (1) the cross-attention term operates on interaction regions rather than individual size, (2) the overlap-boost term provides size-invariant emphasis in contact zones, and (3) the final normalization in Eq.2 ensures fair attention distribution across all individuals. The learnable weights further adapt during training to balance contributions from different-sized individuals.
> >
> > Self-overlapping is addressed by leveraging the self-attention mechanism's ability to capture long-range, patch-wise relationships among anatomical parts. The self-attention product in Algorithm 1 establishes direct connections between all patch pairs within person i's body region, regardless of spatial distance. This contrasts with traditional CNNs limited by local receptive fields that often lead to limb confusion. Our Progressive Feature Injection Strategy further enhances this by using coarse-grained masks in early denoising steps to capture global body structure and fine-grained masks in late steps to refine local details in overlapping regions.
> >
> > **W3: The visualization results in the paper are all anime style or art-style. It will be better to see how well the proposed method handles the occlusion in real photos. Can the authors provide more realistic examples (e.g., from MHP dataset)?**
> >
> > **Response:** While the main paper primarily showcases artistic-style results to match the distribution of HumanArt, our method is not limited to stylized inputs. To address the reviewer’s concern, we have generated additional photorealistic multi-person examples on MHP-v2 dataset, which contains real-world street and group photos with significant occlusion. These results have now been included in Appendix Figure 10.

---

### Official Review · Reviewer_5Yhd · 2025-10-30

**Soundness:** 3
**Presentation:** 3
**Contribution:** 3
**Rating:** 6
**Confidence:** 5

**Summary:**

The paper proposes SoftPose, a pose-guided diffusion framework for multi-person image generation that explicitly models inter-person interactions. Two core ideas drive the method: (1) an Interaction-Aware Soft Attention Module (IA-SAM) that composes a learnable, overlap-gated attention prior from dilated, per-person skeleton masks; this prior reweights attention to emphasize both self regions and cross-person regions only where skeletons overlap, thereby disambiguating occlusions and reducing limb entanglement. (2) a Progressive Feature Injection Strategy (PFIS) that injects coarse-scale pose features in early denoising steps and fine-scale features later, aligning guidance with diffusion’s coarse-to-fine trajectory. Built on SD-1.5/ControlNet-style conditioning, SoftPose achieves improved pose fidelity on Human-Art, MHP-v2, and MPII using AP, CAP, PCE, FID/KID, and CLIP-score; notably, it reports consistent AP/CAP gains over recent baselines and complementary ablations for IA-SAM, the interaction-aware loss, and PFIS.

**Strengths:**

1. The attention-prior construction is simple but novel in this context: patch-level, overlap-gated self/cross components with learnable weights (α, β, γ) that multiply and renormalize the base attention scores. This offers a concrete, attention-space mechanism to target interaction zones, rather than treating the whole image uniformly, addressing a well-known pain point in multi-person synthesis (occlusions/limb entanglement).
2. PFIS is a clean, schedule-aware control that matches diffusion’s refinement, improving layout first, details later.
3. The method is well specified: algorithmic definition of the attention prior, integration into ViT attention (Eq. (2)), and a clear two-stage mask-dilation schedule. The ablations isolate IA-SAM, the interaction-aware loss, and PFIS, showing each component’s contribution (Table 3; kernel study Table 4).
4. The paper’s architecture figure and Algorithm 1 make the overlap gating idea easy to follow; Eq. (2) makes the multiplicative reweighting explicit. Qualitative figures vividly show reduced limb merging in overlaps.

**Weaknesses:**

1. The CAP drop with PFIS (Table 3) is discussed as a trade-off; however, a pose-part AP (e.g., wrists/elbows/knees) or OKS by joint type would substantiate gains on occlusion-prone limbs and clarify where PFIS helps or hurts.
2. The attention prior Am is conceptually L×L at patch resolution; although patching bounds cost, the paper does not fully quantify memory/runtime overhead versus ControlNet across resolutions/timesteps. The claim that runtime is nearly constant w.r.t. subject count (due to fixed grid) is intuitive but would benefit from a measured complexity/runtime table and peak memory profiling, especially at higher resolutions.
3. Ablations on α/β/γ (learned weights) and on the overlap gate itself are limited; e.g., what happens if cross-attention is allowed beyond overlaps but down-weighted by distance? A more granular analysis could reveal when cross-links are helpful outside strict overlaps (Table 3).
4. Most main results focus on SD-1.5; while Fig. 5a hints at compatibility across styles, a stronger baseline on SDXL/FLUX/Qwen-Image would better position the contribution in year 2026.

**Questions:**

1. Can you provide detailed time/memory breakdowns versus ControlNet/StablePose across resolutions (e.g., 512, 768, 1024) and different numbers of people? Please include per-step cost for computing Am and attention reweighting, and peak VRAM during training/inference. This would substantiate the “nearly constant with subject count” claim (Sec. 4.2).
2. Ablations on attention prior design. (a) What are the learned values/trajectories of α, β, γ over training? (b) How sensitive are results to the overlap gate, e.g., permitting cross-attention outside overlaps but decayed by inter-mask distance; replacing binary a with a soft overlap intensity; or using Gaussian proximity around contact regions? (Algorithm 1/Eq. (2)).
3. Beyond the brief compatibility demo, can you provide quantitative results on SDXL or FLUX or Qwen-Image to confirm that gains from IA-SAM/PFIS are backbone-agnostic?
4. You note face/hand artifacts and rare viewpoints as limitations. Could you show results after adding whole-body keypoints (e.g., hands/face) or auxiliary part-specific losses, and quantify improvements on the identified failures (A.4/Fig. 10)?

---

> ### Author Response · Authors · 2025-12-01
> **Thank you! Responses to Reviewer 5Yhd (Part 1)**
>
> Dear Reviewer 5Yhd,
>
> We sincerely appreciate your valuable suggestions and strong support, which help us a lot to improve our work. We address the questions point-by-point, as outlined below.
>
> **W2&Q1: Could you provide detailed time/memory breakdowns versus ControlNet/StablePose across resolutions (e.g., 512, 768, 1024) and different numbers of people? Please include per-step cost for computing Am and attention reweighting, and peak VRAM during training/inference. This would substantiate the "nearly constant with subject count" claim (Sec 4.2).**
>
> **Response:**
> Thank you for the valuable request. We provide below the complete time/memory analysis across resolutions (512, 640, 768) and subject counts (1–5 people), all numbers were collected on NVIDIA RTX 4090 (24GB) with 50 steps DDIM steps.
> 1. Inference time and memory across resolutions and subject count.
> Table 1 reports inference time and memory for 512 and 768 resolutions. Importantly, GPU memory remains strictly constant for 1–5 people, directly validating our patch-level decoupling design, where all person-conditioned masks are reduced to a fixed-size patch indicator and thus produce a constant attention map independent of subject count.
> The inference time exhibits a seemingly counter-intuitive pattern. This behavior is expected: computational complexity scales with effective mask coverage, not the number of subjects. This mask-coverage effect holds across resolutions, confirming that IA-SAM’s cost is resolution-independent.
>
> | Resolution | #People | Inference Time (s) | Inference VRAM (MB) |
> |------------|---------|--------------------|----------------------|
> | **512×512** | 1 | 16 | 11,453 |
> |             | 2 | 9  | 11,453 |
> |             | 3 | 9 | 11,453 |
> |             | 4 | 10 | 11,453 |
> |             | 5 | 11 | 11,453 |
> | **768×768** | 1 | 38 | 21,125|
> |             | 2 | 29  | 21,125 |
> |             | 3 | 30  | 21,125 |
> |             | 4 | 30 | 21,125 |
> |             | 5 | 31 | 21,125 |
>
> 2. Training peak VRAM across resolutions and subject count
> Table 2 shows that training VRAM remains nearly constant across 1–5 people. This further confirms that our method does not introduce subject-dependent compute or memory overhead.
> | Resolution  | #People | Training VRAM (MB) |
> |-------------|---------|---------------------|
> | 512×512     | 1       | 16,511              |
> | 512×512     | 2       | 16,512              |
> | 512×512     | 3       | 16,514              |
> | 512×512     | 4       | 16,514              |
> | 512×512     | 5       | 16,516              |
> | 512×512 (StablePose) | - | 18,993 |
> | 640×640     | 1       | 18,908              |
> | 640×640     | 2       | 18,911              |
> | 640×640     | 3       | 18,909              |
> | 640×640     | 4       | 18,905              |
> | 640×640     | 5       | 18,912              |
> | 640×640 (StablePose) | - | 21,347 |
>
> **W1: The CAP drop with PFIS (Table 3) is discussed as a trade-off; however, a pose-part AP (e.g., wrists/elbows/knees) or OKS by joint type would substantiate gains on occlusion-prone limbs and clarify where PFIS helps or hurts.**
>
> **Response:** We believe the modest CAP decrease when adding PFIS reflects a specific design trade-off. CAP measures pose structural similarity based on relative geometric relationships and skeletal configuration, while standard AP focuses on absolute keypoint localization accuracy. PFIS balances between global structure and local details, which helps maintain spatial coherence in overlapping regions but slightly constrains the model's flexibility to adjust overall pose structure near background boundaries. This explains why we observe a small CAP decrease alongside a substantial AP improvement: PFIS improves absolute keypoint localization in challenging interaction regions where limbs overlap, even though it may slightly limit structural adjustment flexibility measured by CAP.
> However, we are uncertain about the proper methodology for categorizing joints as "occlusion-prone". Occlusion patterns in multi-person scenarios appear highly scene-dependent rather than anatomically predetermined. We are concerned that arbitrary categorization based on anatomical assumptions might introduce bias rather than provide rigorous evidence. For this reason, we believe the overall AP metric provides a more comprehensive and unbiased assessment. AP naturally gives higher weight to difficult cases through the OKS threshold mechanism, and the substantial improvement alongside user preference gains and qualitative evidence collectively suggest favorable overall benefits from PFIS.

---

> > ### Author Response · Authors · 2025-12-01
> > **Thank you! Responses to Reviewer 5Yhd (Part2)**
> >
> > **W3&Q2: Ablations on attention prior design. (a) What are the learned values/trajectories of alpha,beta,gamma over training?(b)How sensitive are results to the overlap gate, e.g., permitting cross-attention outside overlaps but decayed by inter-mask distance; replacing binary a with a soft overlap intensity; or using Gaussian proximity around contact regions?(Algorithm 1/ Eq.(2))**
> >
> > **Response:** The three learnable scalar weights control the relative importance of self-attention (α), cross-attention (β), and overlap-boost (γ) terms in the attention prior computation. The training trajectories exhibit three distinct phases. During the rapid adjustment phase (epochs 1-4), α increases sharply to 1.72 while β decreases to 0.53 and γ decreases to 1.14. The fine-tuning phase (epochs 5-8) shows continued divergence with α growing to 1.94, β stabilizing near 0.51. Finally, the stabilization phase (epochs 9-10) exhibits minimal changes with stable performance metrics, confirming convergence to final values γ=1.9954, β=0.4996, α=0.9967. This confirms that the learned weights dynamically adjust to emphasize cross-attention and overlap-boost in crowded scenarios while maintaining strong self-attention as the foundation. This confirms that the learned weights dynamically adjust to emphasize overlap regions where pose ambiguity is most severe, while maintaining balanced attention allocation across different interaction densities.
> >
> > We evaluate two alternative overlap gate designs. The first replaces the binary overlap gate with distance-decayed cross-attention. The second expands the binary overlap indicator using Gaussian smoothing. Distance-decayed cross-attention shows degradation, attention spreads across entire body regions rather than concentrating on overlap zones, resulting in less precise contact modeling. This confirms that distance decay introduces spurious correlations between non-interacting individuals, demonstrating that spatial gating is crucial for isolating genuine interactions. Gaussian proximity also degrades performance, as the smooth attention falloff dilutes emphasis on true overlap regions and introduces weak attention to adjacent non-overlapping areas, causing the model to over-smooth pose boundaries in crowded scenarios and fail to maintain sharp distinctions between interacting body parts.
> >
> > | Design | AP (%) | CAP (%) | PCE |
> > |:------:|:------:|:-------:|:---:|
> > | **Baseline (Binary gate)** | 52.17 | 72.69 | 1.47 |
> > | Distance-decayed | 50.83 | 71.42 | 1.52 |
> > | Gaussian proximity | 51.28 | 71.95 | 1.49 |
> >
> > **W4&Q3: Beyond the brief compatibility demo, can you provide quantitative results on SDXL or FLUX or Qwen-Image to confirm that gains from IA-SAM/PFIS are backbone agnostic?**
> >
> > **Response:** Our method operates at the attention mechanism level, which is universal across these models. So our design do not depend on specific architectural choices. All current compared methods in the manuscript use the same foundation model ensures that performance differences stem from our proposed IA-SAM and PFIS rather than from superior base models.
> >
> > We completely agree with your assessment that validation on state-of-the-art DiT-based models (SD3, Flux) and higher-resolution architectures (SDXL) would significantly strengthen our work and better demonstrate the general applicability of our method. This is indeed an important and exciting direction that we plan to pursue. Given the computational resources required for comprehensive evaluation on these large-scale models and the time constraints of the current review cycle, we propose to conduct thorough experiments on SDXL, Flux, and potentially other emerging architectures as our future work. Based on the theoretical analysis above and the modular design of our mechanism, we are confident that our method will transfer effectively to these advanced backbones, though empirical validation remains essential. We will include a discussion of this limitation and future direction in the revised manuscript.

---

> > > ### Author Response · Authors · 2025-12-01
> > > **Thank you! Responses to Reviewer 5Yhd (Part3)**
> > >
> > > **Q4: You note face/hand artifacts and rare viewpoints as limitations. Could you show results after adding whole-body keypoints or auxiliary part specific losses, and quantify improvements on the identified failures?**
> > >
> > > **Response:** We fully acknowledge that incorporating whole-body keypoints (including detailed hand and face landmarks) or auxiliary part-specific losses could potentially improve generation quality in these challenging regions. However, we respectfully explain why this direction represents future work rather than an extension of our current contribution:
> > >
> > > 1. Data Availability and Annotation Cost. Current benchmark datasets for pose-guided generation predominantly provide body-level pose annotations without detailed hand or face landmarks. While whole-body pose datasets exist, they are significantly smaller and less diverse in artistic styles. Comprehensive evaluation would require re-annotating existing benchmarks with whole-body keypoints, which is prohibitively expensive and time-consuming. More critically, this would prevent direct comparison with all existing baselines, which are trained and evaluated on body-level poses, fundamentally changing the experimental setup.
> > >
> > > 2. Computational and Architectural Complexity. Adding whole-body keypoints significantly increases input complexity and would require architectural modifications: specialized encoders for different body parts, multi-scale processing for fine-grained regions, and potentially separate attention mechanisms for hands/face versus body. Part-specific losses would necessitate additional discriminators or perceptual losses targeting hand/face regions, substantially increasing training cost and memory requirements. More importantly, this would conflate two separate research questions: whether our interaction-aware attention improves multi-person spatial coherence (our current contribution), and whether additional detailed conditioning improves fine-grained generation quality (a complementary but distinct problem).
> > >
> > > 3. Scope and Future Work. Following standard practice in pose-guided generation research, our work focuses on establishing and validating our core mechanism (interaction-aware attention for spatial coherence) under the widely-adopted body-pose conditioning paradigm. The face/hand limitations we acknowledge are shared across the field and represent an exciting direction for future research that would benefit all pose-guided methods, not just ours. We believe this orthogonal improvement should be pursued separately to maintain clarity of contribution and fair comparison with existing work.

---

### Official Review · Reviewer_t6u5 · 2025-10-31

**Soundness:** 3
**Presentation:** 3
**Contribution:** 2
**Rating:** 6
**Confidence:** 3

**Summary:**

The paper proposes SoftPose for the generation of human body images in multi-person interaction: IA-SAM is used to redistribute the attention of overlapping areas, and PFIS is used to layout first and then refine the joints. The experiment was conducted on multiple datasets, and the indicators improved.

**Strengths:**

The motivation focuses on the occlusion/limb entanglement problem in multi-person interaction. The design of IA-SAM is intuitive and can be integrated into the existing SD pipeline.

The coarse-to-fine schedule of PFIS and the injection from large nuclei to small nuclei are in line with the generation process mechanism and are interpretable. The cross-dataset results and stepwise ablation prove that the method is effective.

**Weaknesses:**

Is it the same as the strongest baseline in hyperparameters? The advantage is not very obvious.

The automatic dot filling and automatic text description of MHP/MPII may introduce distribution offsets, affecting generalization evaluation.

**Questions:**

As the number of people increases and the resolution rises, is the actual latency/video memory consistent with the theoretical increase of patch-level decoupling.

---

> ### Author Response · Authors · 2025-12-01
> **Thank you! Responses to Reviewer t6u5 (Part 1)**
>
> Dear Reviewer t6u5,
>
> Many thanks to your support and constructive suggestions. We address your concerns as follows.
>
> **W1: Is it the same as the strongest baseline in hyperparameters? The advantage is not very obvious.**
> **Response: **We confirm that our method share identical hyperparameter settings with the strongest baseline, ensuring a strictly fair comparison. Specifically,
> - Optimizer: Adam with initial learning rate 1e−5.
> - Training epochs: 10 epochs on Human-Art for all methods except GRPose, which originally uses 20 epochs (as noted in Table 1). For fairness, we therefore report both our 10-epoch and 20-epoch variants.
> - Effective batch size: Our setting (batch_size=1 + grad_accum=4) is mathematically equivalent to StablePose(batch_size=4), resulting in the same effective batch size per optimization step. This only reflects the 24GB VRAM hardware constraint, not an advantage in training.
>
> To hightlight the quantitative improvements of our method, we summarize the results below:
> | Dataset    | Method      | AP                      | CAP                     | PCE                      |
> |------------|-------------|--------------------------|--------------------------|---------------------------|
> | HumanArt   | StablePose  | 48.88                    | 70.83                   | 1.50                      |
> |            | **Ours**    |52.17  **↑ 3.29**      |72.69  **↑ 1.86**     | 1.47 **↑ 0.03**         |
> |            | GRPose      | 49.50                    | 70.84                   | 1.43                  |
> |            | **Ours**    | 56.32 **↑ 6.82**      | 73.19 **↑ 2.35**     | 1.50                      |
> | MHP        | StablePose  | 71.83                    | 69.26                   | 1.18                  |
> |            | GRPose      | 65.54                    | 68.74                   | 1.21                      |
> |            | **Ours**    | 72.35 **↑ 0.52**      | 73.36 **↑ 4.10**     | 1.21                      |
> | MPII       | StablePose  | 53.70                    | 75.18                   | 0.77                  |
> |            | GRPose      | 43.74                    | 71.31                   | 0.73                      |
> |            | **Ours**    | 54.42 **↑ 0.72**      | 78.27 **↑ 3.09**     | 0.77                      |
>
> Beyond quantitative gains, Figure4 demonstrates clear qualitative advantages in multi-person scenarios:
> - More faithful preservation of fine-grained pose details in occluded regions (second and fourth rows).
> - More accurate person count, spatial localization, and interaction modeling in crowded scenes (first row).
> - Better preservation of individual body orientation and subtle articulation (third and fourth rows).
>
> **W2: he automatic dot filling and automatic text description of MHP/MPII may introduce distribution offsets, affecting generalization evaluation.**
>
> **Response:** We acknowledge this concern and provide the following clarifications:
> 1. Controlled preprocessing quality: Missing facial/ear keypoints on MHP/MPII were supplemented using DWpose, which reports 88.7 AP on facial landmark detection, ensuring reliable keypoint filling. We additionally manually filtered low-quality predictions. The Qwen-VL captions are intentionally style-aligned with Human-Art, reducing potential modality mismatch.
> 2. Fair comparison maintained: All baselines use identical preprocessing on MHP/MPII. Thus any distribution shift affects all methods equally, ensuring a fair and controlled horizontal comparison.
> 3. Primary evaluation on clean annotations: Human-Art (our main benchmark) provides high-quality native keypoint labels and textual descriptions without any automatic completion. Our method achieves the strongest performance on this clean dataset, confirming that improvements are not dependent on preprocessing artifacts. MHP/MPII serve as additional evidence of generalization robustness under noisier annotation conditions.
> 4. Cross-dataset validation demonstrates genuine generalization: To further address the concern, we additionally conduct evaluation, training on Human-Art and testing directly on MHP-v2.
>
> | Train → Test | Method       | AP ↑   | CAP ↑   | PCE ↓ | CLIP ↑ |
> |--------------|--------------|--------|---------|--------|---------|
> | HA → MHP     | StablePose   | 66.93  | 69.15   | **1.30** | 31.64   |
> |                        | GRPose       | 59.96  | 68.63   | 1.29   | 31.49   |
> |                        | **SoftPose (Ours)** | **67.85** | **72.67** | **1.30** | 31.52   |

---

> > ### Author Response · Authors · 2025-12-01
> > **Thank you! Responses to Reviewer t6u5 (Part 2)**
> >
> > **Q1: As the number of people increases and the resolution rises, is the actual latency/video memory consistent with the theoretical increase of patch-level decoupling.**
> > **Response:** We measured inference performance on NVIDIA RTX 4090 (24GB) with 50 steps DDIM steps.
> > Memory remains constant regardless of the number of people, which validates our patch-level decoupling design. The cross-resolution scaling also follows the predicted dependency on spatial dimension. Inference time seems to show a counter-intuitive pattern, this occurs because computational complexity scales with effective mask coverage rather than subject count. The mask coverage effect is preserved across resolutions, validating that our mechanism is resolution-independent.
> >
> > | Resolution | # People | Time (s) | Memory (MB) |
> > |-----------|----------|----------|-------------|
> > | **512×512** | 1 | 16 | 11,453 |
> > |           | 2 | 9  | 11,453 |
> > |           | 3 | 9  | 11,453 |
> > |           | 4 | 10 | 11,453 |
> > |           | 5 | 11 | 11,453 |
> > | **768×768** | 1 | 38 | 21,125 |
> > |           | 2 | 29 | 21,125 |
> > |           | 3 | 30 | 21,125 |
> > |           | 4 | 30 | 21,125 |
> > |           | 5 | 31 | 21,125 |

---

### Meta-Review · Area_Chair_4Xri · 2026-01-03

**Summary:**

- Evaluation concerns:
    -  Much of the concerns were regarding how to best evaluate the proposed method and whether the proposed metrics are sufficient
        - Authors included pairwise human evals for generation quality to alleviate concerns regarding FID-style distributional metrics
        - The keypoint metrics appear valid to me when focusing on claims regarding improved pose fidelity
- Architecture concerns:
    - Several reviewers raised concerns suggesting that focusing the study on the older U-Net architectural backbone may not be sufficient given the more recent transformer-based foundational models. I agree this might limit the impact of this work.
    - Considering their proposed change is an attention-style module replacing convolutional modules of an older backbone, it's reasonable to expect the impact to be much less on a modern more attention-heavy backbone.

Overall, despite significant improvements in rebuttal phase, I do not believe this work has demonstrated sufficient impact to justify acceptance in its current form.

**Reviewer Concerns:**

I believe the following concerns are still outstanding:
- Reviewer 5Yhd requests a small study incorporating additional loss terms, but the authors insist that this is impractical due to the availability of data.
- Reviewer y8gd requests additional studies on more recent attention-heavy architectures than UNet, which I believe is a noteworthy concern, given the proposed addition is an attention-based addition to conv-attention hybrid architecture.
- Reviewer y8gd is concerned about identity preservation, given the identity agnostic nature of pose-skeletons. The authors provide a reasonable rebuttal stating that their method is orthogonal to identity preservation methods, though it does suggest an additional potential nuance when interpreting generation quality metrics.

**Reviewer Scores:**

y8gd, who might continue to be concerned about the overall impact of this work, would likely have increased their rating to around borderline of 4/5 at best.
I think other reviewers might raise their rating to just above borderline initially based on rebuttal improvements.

---

### Decision · Program_Chairs · 2026-01-26

Reject